# Review of Studies Regarding Assessment of Families Where Children Are at Risk of Harm Due to Parental Substance Misuse

**DOI:** 10.3390/ijerph22040612

**Published:** 2025-04-14

**Authors:** Richard D. Tustin

**Affiliations:** Adelaide Psychological Services, Adelaide 5045, Australia; don@psychadelaide.com.au

**Keywords:** drugs, family violence, psychoactive substances, heavy substance use, health and safety promotion, early intervention therapy, illicit substance, cumulative harm, parental risk factors, level of risk, severity of substance use, threshold, parenting capacity, multiple complex needs, parental mental health, coordination of care, confidentiality, objective assessment, transdiagnostic therapy

## Abstract

Questions arise about links between the use of substances and violence, especially when parents misuse substances and there is potential to expose children to family violence. Background. The review has four aims: identify research into the harmful impacts of parental substance use on children; identify policies in Australia about the risks from parental substance misuse; identify interventions to manage unsafe parental substance use; and review practices involving confidentiality and collaboration when a parent has multiple complex needs. Method. The paper provides a rapid review of the literature, linking parental substance misuse, family violence, and parenting capacity and covering both policies and empirical evidence. Results. The main finding is that parental substance misuse does affect parenting capacity and is associated with family violence. The concept of the cumulative risk of harm to vulnerable children is supported by research but is not yet implemented in policy. Reports indicate that some parents who misuse substances have multiple complex needs including comorbid mental health issues, domestic violence, and difficulty in managing their children’s behavior. Early intervention therapies designed to address this cluster of issues are reviewed. Conclusion. There is a need to establish objective assessment instruments that are relevant to the cohort of parents who misuse substances and engage in family violence and to improve policy to ensure vulnerable children and families in which parents misuse substances are referred to appropriate therapies.

## 1. Introduction

There is growing concern about the impacts of domestic and family violence and the causes of family violence. There is concern that frequent use of psychoactive substances may be linked to violence, especially when aggression occurs in a family home and children are exposed to aggressive behavior by a parent. The question of how to manage family violence associated with the frequent use of psychoactive substances is viewed as a complex policy topic.

Policy makers in Australia report that children exposed to family violence first come to notice when a notification is made to the child welfare service that a child is at risk of harm from their own parent [1]. However, policy makers are uncertain about how to respond to this scenario [2].

In 2024, the State Government of South Australia established the Royal Commission into Domestic, Family and Sexual Violence. The terms of reference proposed the continuation of a whole government approach that was introduced following an earlier Royal Commission [3] that focused on how to facilitate the delivery of preventive and early intervention services to families where children are vulnerable. The current Royal Commission was established following reports that a woman is killed by an intimate partner in Australia every 11 days; 39% of Australian women have experienced domestic violence; and 29% of women have experienced violence from a cohabiting partner.

There are further reports that children are at risk of serious harm and death from family violence, including following notification that a child is at risk of harm to the child welfare department. An annual report to Parliament by the South Australian Child Death and Serious Injury Review Committee stated that in the 16-year period of 2005–2020, 473 children had died within three years of being notified to the department for child welfare as being at risk of harm, which averaged 29.6 deaths of children per year in the state [4]. The report considered the life circumstances of parents whose child died and identified contributing factors to the deaths of children to include parental violence, parental mental and physical health conditions, and a parent having themselves been raised in out-of-home care. While parental misuse of substances was not identified as a risk factor for children in this report, it has been identified as a risk factor in other reports.

A report by the Australian Institute of Criminology into filicide in the 12-year period between 2001 and 2012 found that of the children who were killed, 84% were killed by a parent, with adult males and females contributing about equally [5]. Police reports found that parents had consumed substances at the time of the filicide in 23% of cases. Other comorbidities were domestic violence in 30% of cases, parental mental illness in 32% of cases, and criminal convictions in 43% of cases.

One aim of initiatives in Australia is to reduce gaps between policies and clinical practices to ensure that children at risk of harm receive appropriate protection, therapy, and support. This review sets out to identify therapy services that can be provided as early intervention for children at risk due to parental substance misuse.

### Objectives

This paper proposes that the government’s aim to provide early intervention therapies for vulnerable children relies on identifying an objective assessment instrument that identifies families where children are at risk of harm and distinguishes between families who require early intervention therapy and families where children need to be removed from parental care to protect the child.

This paper distinguishes between policies and practices. The paper considers policy and empirical evidence on the following topics: (a) associations between parental use of psychoactive substances and harm to children; (b) broad government policies; (c) information about risk factors/predictors of harm to children; (d) information about the scale of the problem; (e) information about the efficacy of therapies for people who misuse substances; (f) collaboration between health, welfare, and legal services; (g) issues of privacy and confidentiality regarding parents who misuse substances; and (h) proposals about how services might be delivered collaboratively when parents are assessed as having complex needs and require input from different providers.

## 2. Methods

### Review Protocol

A search was conducted into the literature about three issues: (a) empirical studies about risk factors for children associated with parental substance misuse; (b) policy principles about children who are exposed to harm due to parental substance misuse; and (c) evidence about treatments of parental substance misuse.

Databases were searched using Google Scholar for articles published in English between the dates of 2000 and 2024. Articles were accessed using the following keywords. Keywords for risk to children were drugs, family violence, psychoactive substances, heavy substance use, illicit substance, cumulative harm, severity of substance use, parental risk factors, risk level, multiple complex needs, threshold, meta-analysis, and health and safety promotion. Keywords for policy principles were Australia, objective assessment, parenting capacity, abstinence, outcome measure, coordination of care, privacy, and collaboration. The keywords for therapies were early intervention therapy and transdiagnostic care.

The literature about policy principles was restricted to policies in Australia. International empirical studies about risk factors and therapeutic interventions were included. The Australian Commonwealth Government assigned the role of monitoring the efficacy of child welfare practices to the Australian Institute of Health and Welfare (AIHW), which collects and publishes data annually on a set of indicators. This paper cites data from the AIHW on relevant topics. An analysis was conducted on the consistency of the use of instruments in research to measure constructs.

## 3. Results

Information about policy principles and empirical evaluations are discussed together.

### 3.1. A–Associations Between Parental Substance Misuse and Harm to Children

#### Policy and Educational Practices

Bromfield, Sutherland, and Miller [6] published a document about how the Victorian Department of Human Services views the impacts of parental substance use on parenting practices in families with multiple and complex needs. The document proposes that both intoxication and withdrawal impact a parent’s ability to perform basic parenting tasks including maintaining routines, supervising children, and meeting a child’s emotional needs as parenting practices become inconsistent when a parent is intoxicated. Predicted impacts on children include a high risk of neglect, risk of physical abuse from a parent, and risk of sexual abuse.

Concerns of child welfare services are based on generally available information about the impacts of psychoactive substances including alcohol, cannabis, and illicit substances. Early information sheets provided by child welfare services provide the following views about the use of psychoactive substances by parents. Frequent use of psychoactive substances by parents interferes with their thinking processes and their parenting practices and contributes to the use of extreme parenting styles as substance use alters the parent’s state of consciousness and ability to regulate their emotions. When intoxicated, some parents use more authoritarian parenting styles and expose their child to harsh and unreasonable punishments and to the parent’s high level of emotionality. Other parents use under-involved parenting styles when intoxicated, exposing their children to emotional and physical neglect. Children whose parents use high levels of substances are exposed to inconsistent parenting practices. Many children of substance-abusing parents face several adverse childhood events during their formative years of 2–12 years. Extreme parenting styles are associated with an increased prevalence of a range of emotional and behavioral problems in children.

### 3.2. Evidence About the Impacts of Parental Substance Misuse on Children

Chaffin et al. [7] conducted a two-year follow-up study of more than 7000 parents with no prior history of child maltreatment and found that the presence of parental substance misuse increased the odds of child abuse by 2.9 and of child neglect by 3.24.

Gruenert [8] documented the experiences of 48 Australian children whose parent attended treatment for substance dependence. Gruenert found that over half of the children had been adversely affected by their parent’s substance misuse due to their exposure to family violence, family breakdown, police raids on the home, and children’s play areas being searched for drugs. The study found that a third of children had been exposed to experiences of finding their parents unconscious and being exposed to other dangerous situations. In some cases, children were neglected. Assessments found that a quarter of children scored in the clinical range for being emotionally disturbed. Child welfare services had been involved with 40% of the children. Children who were most severely affected lived in a single-parent family. The report described the children as ‘nobody’s client’ as no specific agency provided therapy for affected children. The study made the point that parental misuse of substances can be associated with a range of adverse experiences for children.

Conners et al. [9] studied 4084 children in the USA whose mothers were admitted to publicly funded residential substance abuse treatment programs for pregnant and parenting women and found their children had increased vulnerability for physical, academic, and social-emotional problems, and that many children required long-term supportive services.

Fergusson, Boden, and Horwood [10] reported a prospective study over 25 years involving the health, development, and adjustment of a cohort of 1265 children born in New Zealand. They found that parental use of illicit substances predicted illicit substance use by their children.

Bromfield et al. [2] analyzed families where children were placed in out-of-home care in South Australia and found that of the children who entered care due to parental substance abuse, 69% of parents also experienced domestic violence and 65% of parents had a mental health problem. The authors identified families who presented with more than one risk factor as having multiple and complex problems and noted that these families had become the primary client group of the child protection service. Bromfield et al. cited a report by the Special Commission of Inquiry into Child Protection in New South Wales involving 302,977 child protection reports in New South Wales during 2007–2008 that found that the frequencies of the three most prominent parental risk factors were domestic violence in 31% of cases, drug and alcohol problems in 20% of cases, and mental health conditions in 14% of cases ([11], p. 130). Bromfield et al. cited data from Victoria in 2001–2002 showing that the four most frequent concerning characteristics in parents who were investigated were domestic violence in 40% of cases, illicit substance abuse in 25% of cases, alcohol abuse in 21% of cases, and parental psychiatric illness in 15% of cases.

Data summarized by Bromfield et al. indicate that some notified parents experience only substance misuse; some parents experience dual conditions of substance misuse and a mental illness; and some parents experience three conditions involving substance misuse, mental illness, and family violence.

Jeffreys et al. [12] reported a study that was commissioned by the South Australian Child Welfare Department. Jeffreys and colleagues studied families where children were placed into out-of-home care in 2006 due to substantiated parental misuse of substances, and this involved 40% of all children in care. The substances most used by parents were alcohol (77% of parents), cannabis (53%), amphetamines (50%), heroin (12%), and prescribed medications (11%). Regarding the frequency of use of substances, they found that alcohol was used daily by 27% of parents, cannabis was used daily by 37% of parents, and amphetamines were used daily by 47% of parents.

The Jeffreys study found that if a parent misused a substance, there was an increased likelihood that the family experienced other difficulties. The following percentages of families were reported to be experiencing other difficulties: 69% of parents experienced domestic violence, 65% experienced a parental mental health issue, 29% had financial difficulties, 28% had issues of homelessness or transient accommodation, and 25% had issues of parental incarceration. Families who misused substances had a median of five problems, with some families having ten problems. These statistics show that parental substance misuse is associated with multiple complex problems in some families where a notification was made to a child welfare service.

Jeffreys’ Table 8 shows that the types of abuse experienced by children whose parent misused substances were as follows: 52% of children were exposed to domestic violence, 57% were assessed as being neglected, 54% had unstable living arrangements, and 32% were exposed to drug use and dealing. The table identifies four common risk factors for children.

Jeffreys also found that 25% of children had been placed into care although no evidence of maltreatment of the child was provided. The study found that a quarter of the children placed into out-of-home care were placed into care while the family received assistance to manage a short-term family crisis, to allow the family time to recover rather than simply to protect children from harm (page 30). The study found that for some parents, placement of their child into care provided a parent with space to ‘get their act together’ while caseworkers provided support, information, advocacy, referrals, and linkages that helped parents to make and sustain positive changes. This was described as a therapeutic approach.

The Jeffreys study also found that some parents were apathetic about their children being returned to their parent’s care, were hostile to workers, minimized child protection concerns, and appeared to lack the capacity to change. The approach used by case workers with children of these parents was described as risk management to protect the child.

The Jeffreys study included a case file review that found that child welfare staff referred parents to the following separate support services: substance abuse services in 58% of cases, parenting education in 45% of cases, general counseling in 33% of cases, adult community mental health in 29% of cases, child mental health in 26% of cases, housing in 24% of cases, domestic violence services in 17% of cases, financial counseling in 18% of cases, and child care in 16% of cases. The mean number of services each family was referred to was 5.3.

The case review by Jeffreys found that when child welfare services referred cases for drug and alcohol intervention, they referred the most serious cases rather than using an early intervention therapy approach. The file analysis found that only in 41% of referred cases recorded parents received a drug and alcohol intervention and that 42% of parents disengaged prematurely from a service. The Jeffreys study found that most parents with a substance abuse disorder had not contacted a drug and alcohol service provider prior to their child entering care, apart from 30% of parents who were engaged in a methadone maintenance program on referral by their general practitioner. Jeffreys and colleagues concluded that substance-misusing parents are a ‘difficult to reach’ population.

The Jeffreys study examined whether parents who were referred to drug and alcohol services received an intervention that was child-focused and concluded that 80% of families who engaged with a rehabilitation service did receive a child-focused service. The Jeffreys study assessed whether parents who engaged with rehabilitation received multi-systemic interventions that aimed to address the multiple needs of the family and concluded that 58% of families did receive a service that was individually tailored to target all needs of the family. The study examined whether efficacy of interventions was individually evaluated and concluded that individualized evaluations and adjustments were made in 58% of cases and that inter-agency collaboration was achieved in 73% of these cases.

The Jeffreys study noted that the child welfare system can become a gateway to treatment for drug and alcohol issues for families where children are vulnerable if adequate assessment instruments are used to guide referrals. A policy encouraging child welfare services to refer families where children are vulnerable would enhance the delivery of early intervention therapies to families.

Doidge et al. [13] interviewed adults from a population-based cohort of 2443 Australians about their childhood history to identify risk factors for maltreatment when they were a child. Their study identified parental substance misuse as being a risk factor for children, along with economic disadvantage and social instability.

A report by Wright and colleagues [14] cited Australian research indicating that domestic and family violence often co-occur with parental alcohol and other drug issues and mental health issues, when notifications are made regarding child abuse or neglect, and that the co-existence of these three risk factors often precipitates involvement with the child protection service.

Many child welfare services now believe that frequent parental use of psychoactive substances is a marker for the presence of other risk factors for children [15,16,17,18]. There is a belief that parents who use substances heavily experience more difficulty in managing parenting tasks and everyday stressors, especially if a parent experiences socioeconomic disadvantages. Some agencies view all parents who misuse substances as experiencing disability and propose that all parents who misuse substances be offered a holistic wraparound service that provides a package of many types of support. However, a meta-analysis of 24 studies where mothers had substance abuse problems by Hatzis et al. [19] found that the group was not homogeneous. A review of maternal sensitivity to their child’s cues and children’s responsiveness found significant heterogeneity between mothers, indicating that separate assessments need to be made of maternal substance use and maternal sensitivity to cues.

Kuppens et al. [20] conducted a meta-analysis on the impacts on children of parental use of alcohol, tobacco, and other drugs. They reviewed 56 studies and found 220 dependent effect sizes. A multilevel random-effects model revealed a statistically significant, small detriment to child wellbeing from parental substance abuse over time (r = 0.15), and moderator analyses demonstrated that the effect was more pronounced for parental drug use (r = 0.25) compared with alcohol use (r = 0.13), tobacco use (r = 0.13), and alcohol use disorder (r = 0.14). They concluded from their data that there is a need for future studies to better capture the effect of parental psychoactive substance abuse on the full breadth of childhood wellbeing outcomes and to integrate substance abuse into models that specify the precise conditions under which parental actions determine child wellbeing.

Overall, empirical research shows that educational practices followed by State child welfare agencies are consistent with empirical research. However, research also finds that parents who use substances are not a homogeneous group, as some parents have a single issue of substance misuse while other parents have many issues and present with multiple and complex problems. This finding indicates there is a need for an objective assessment instrument to clarify the needs of each family where a parent uses substances to support the effective delivery of services to this cohort and meet the individual needs of each family. Research indicates that the most common concerns regarding parents who misuse substances involve comorbid parental mental health issues, family violence, and difficulty in managing children’s development and behavior. These families are labeled as having multiple complex needs and as being difficult to reach.

The Jeffreys study found that while the child welfare department can refer vulnerable families to relevant early intervention services after receipt of a notification, the department does not perform this role efficiently, perhaps due to a lack of an agreed objective assessment instrument. Data about the impacts of parental substance misuse on children have policy implications, indicating there is a need for more effective methods to refer families where children are exposed to a moderate level of risk to early intervention therapies.

### 3.3. B–Broad Government Policies

The Government of the State of South Australia and the Australian Commonwealth Government have implemented policies that are relevant to providing support for parents who use substances. Eight government-initiated reviews are summarized.

#### 3.3.1. i–Loxley Review

A review commissioned by the Australian Government about steps to prevent the harmful use of substances was reported by Loxley et al. [16]. The review found that government initiatives had previously focused on educating the general community about risk factors, reviewing legislative measures, and modifying medical interventions. Based on the evidence reviewed, the Loxley group recommended continuing a whole population approach using universal strategies, while adding further levels of intervention, including targeted interventions for cohorts in the population who exhibit moderate levels of risk for harming children.

Parents were identified in the Loxley review as an important group to receive targeted interventions, and parents who used illicit drugs were identified as an important cohort for targeted interventions. The review identified effective interventions for vulnerable families, including education about the adverse impacts of parental drug use on children and home-based and group skill training for both parents and children, with case management for some families. The review reported that evaluations of moderately intensive family intervention programs had demonstrated positive improvements over one to two years in child behavior problems. Targeted programs reduced rates of substantiated child abuse and neglect. The review recommended that to maximize effectiveness, intervention strategies should be provided early in the developmental pathway of the disorder and aim to enhance protective factors and reduce risk factors. The review recommended an investment in drug treatment programs for parents to ensure healthy child development.

The Loxley review discussed the use of cannabis, noting that cannabis was the most widely used illicit drug in Australia with around 10% of adults being regular heavy users of cannabis and at risk of long-term health consequences including dependence. Acute harms from cannabis use include anxiety, dysphoria, panic, and paranoia. Long-term harmful impacts of prolonged cannabis use on mental health include negative impacts on attention, memory, and concentration, producing modest impairments in cognitive functioning.

The review found that heavy ‘binges’ on amphetamine-type drugs were associated with reckless and aggressive behavior and, when sustained over days, may precipitate psychosis. This finding indicates that there is a need for clearer policy regarding amphetamine-type drugs.

The review found that the negative effects of illicit drug use were most clearly predicted by the cumulative number of elevated risk factors, rather than by any specific risk factor. The review also noted that judicial processes have the capacity to divert illicit drug users to effective intervention programs at an early stage in their drug use.

#### 3.3.2. Empirical Evidence About Early Intervention

Dawe and Harnett [21] reported an early intervention program called Parent under Pressure (PUP) that was delivered in one state of Australia. The PUP program is summarized below in Section 3.3 and Section 3.11. Dawe and Harnett recommended that parents who misuse substances be referred to a therapy program, provided for about six months independently of the child welfare service and that clinicians provide a treatment report to an authority about the parent’s participation in the program. No research was found into whether this recommendation has been adopted or about the impacts of the recommendations.

#### 3.3.3. ii–Nyland Royal Commission 2016

The South Australian Government arranged a Royal Commission to investigate the operations of its child protection system, resulting in the Nyland Report [1]. Justice Nyland presented the findings and made recommendations.

Nyland pointed out that the child protection system was an outmoded model as it focused on recording specific incidents of mistreatment of children, and it did not identify risks that cumulate over time. Further, tools used by the department to assess risk were of poor quality, had low inter-rater reliability, were not used by many departmental offices, and were not integrated with decision-making about therapy interventions (pp. 193–194). Nyland found that some departmental assessors showed excessive optimism that parents could change ingrained practices involving substance use (pp. 191–192). Nyland recommended that assessments for court purposes be outsourced to independent expert assessors (p. 16).

Nyland discussed referrals by departmental staff for family-oriented therapy. Nyland found that a high proportion of 61% of notifications were screened out as not requiring any form of investigation or intervention (p. 16), and this resulted in many allegations being recorded but not substantiated independently of the notifier. The practice of screening out notifications resulted in many children receiving no assistance (pp. 196–197), representing missed opportunities for families to be referred for early intervention therapy. Tools used by the department did not identify thresholds to identify families who required a differential response to departmental involvement (pp. 162, 195).

Nyland found that departmental practitioners rarely referred families to external therapists, and there was little evidence of practitioners following recommendations of external providers who were involved with a child (p. 193). When a practitioner referred a family to an external therapist, the practitioner often closed the file and did not follow up on progress (pp. 193–194). Nyland noted that the departmental practice of referring families with multiple needs to multiple providers resulted in services to vulnerable families not being coordinated (p. 157). These findings indicate a need to improve departmental practices regarding parents who misuse substances.

Nyland recommended that children assessed at a moderate risk of harm be referred for therapy interventions (recommendation 163); that therapy be provided independently of the department (p. 17); that families be referred to agencies that are capable of providing therapy services that match the needs of individual families, especially when a family has multiple complex needs (recommendations 66 and 85); and that assessors use structured assessment instruments (p. 201).

Nyland found that the use of expert assessors who are funded by the department can result in assessments that are partisan and that include advocacy, rather than being balanced (p. 201). This finding supports the use of assessors who are independent of stakeholders and who demonstrate their objectivity by using validated assessment instruments.

Nyland discussed communication between government departments. She noted that departments had been established to address single issues, and this has resulted in a situation where departments operate in isolation from one another with little formalized communication regarding the treatment of individual citizens in what she called a “siloed” approach (p. 160). The siloed approach resulted in families with multiple needs being referred to several agencies for services, without any clinical coordinator being appointed to ensure that service delivery was staged and collaborative (p. 160).

Nyland drew attention to the range of government departments that can become involved when a parent misuses substances, including a drug and alcohol department that manages addictions; a health department that treats people with physical and mental illnesses; a child welfare department that manages children at risk of harm; and disability support services that assist people who have impairments due to an ongoing illness. Nyland drew attention to a need for policies that promote the sharing of information and the coordination of delivery of early intervention services.

The Nyland Commission identified several topics that have implications for policy and warrant further research. However, no independent empirical studies were identified that examined the implementation of the Nyland recommendations.

#### 3.3.4. iii–National Framework for Protecting Australia’s Children 2009

In 2009 Australia adopted the National Framework for Protecting Australia’s Children, which provides a policy approach to guide the delivery of interventions to meet the needs of families where children are vulnerable. The framework proposes the use of four categories of families and four linked categories of intervention. The four categories are as follows: competent parents who are eligible for universal preventive supports that are available without restriction, including preventive education for the whole community; people in vulnerable families who are eligible for indicated early intervention services with restricted access; at-risk children who are eligible for focused intervention programs that are restricted to this cohort; and children who are at an unacceptable risk of harm and who are placed into statutory care or out-of-home care.

The National Framework was updated and renamed Safe and Supported 2021–2031. Safe and Supported proposes a whole-of-government approach to improve liaisons between departments based on a recognition that some parents who misuse substances have many complex needs. Safe and Supported identified a need to strengthen the interface and integration of services involving drugs and alcohol, domestic and family violence, mental health, disability, education, justice, housing, and employment services. The policy seeks to develop multidisciplinary models of care for families who experience multiple and complex needs.

#### 3.3.5. Empirical Evidence About Implementing the National Framework

No independent studies were identified that assessed implementation of the National Framework for Protecting Australia’s Children. The National Framework has ongoing implications for policy development and research.

#### 3.3.6. iv–National Children’s Mental Health and Wellbeing Strategy 2021

The Australian Government addressed dilemmas around eligibility criteria and promoted access to early intervention therapies by adopting the National Children’s Mental Health and Wellbeing Strategy 2021, which introduced a wellbeing continuum. The wellbeing continuum describes a child’s mental health on a four-step continuum, with the following steps: *well,* where a child experiences positive mental health; *coping,* where a child experiences challenges they are equipped to manage; *struggling,* where a child experiences challenges they are not managing effectively and that they require focused support to manage; and *unwell,* where a child meets the criteria for having a mental illness and requires treatment. This policy encourages the provision of early intervention therapies for families when a child is struggling, instead of waiting for the child to deteriorate and become unwell.

#### 3.3.7. Empirical Evidence About the National Children’s Mental Health and Wellbeing Strategy

No independent studies were identified that evaluated the implementation of the National Children’s Mental Health and Wellbeing Strategy. The Children’s Wellbeing Strategy has implications for ongoing policy development and research.

#### 3.3.8. v–National Standards for Out-of-Home Care 2011

The Australian Government introduced the National Standards for out-of-home care in 2011, which require state departments to provide information about 13 indicators. Standard 5 is ‘Children and young people have their physical, developmental, psychosocial and mental health needs assessed and attended to in a timely way’. Standard 10 is ‘Children and young people in care are supported to develop their identity, safely and appropriately, through contact with their families, friends, culture, spiritual sources and communities and have their life history recorded as they grow up’.

#### 3.3.9. Empirical Evidence About Implementing the National Out-of-Home Care Standards

The AIHW [22] reported that each year, about 3% of all Australian children aged 0–17 years were abused and subsequently assisted by child protection systems. The frequencies of each type of substantiated abuse of children in child protection systems in 2019–2020 were as follows: emotional abuse in 54% of cases, neglect in 22% of cases, physical abuse in 14% of cases, and sexual abuse in 9% of cases. The AIHW reported that the number of children in out-of-home care in Australia increased by 44% between 1999 and 2009. The proportion of children in out-of-home care differed across jurisdictions, ranging from 4.3 per 1000 in one state to 9.4 per 1000 in another state. These data indicate a growing problem and a need to update policies to address emerging issues.

AIHW [23] reported that the purpose of investigations by child welfare is to determine whether to provide a child with a child welfare service. Of children who did receive a child welfare service, 68% received an investigation of the alleged abuse or neglect, with about 41% of notifications being substantiated and the child receiving a further service of an out-of-home placement. Of notified children, 65% had been the subject of a previous notification. This data illustrates that child welfare services did not focus on ensuring that early intervention therapies were one of their roles.

The AIHW Out-of-Home Care Survey National Dataset [24] provides information about the views of children aged 8–17 years in out-of-home (OOH) care, collected by the state/territory departments responsible for child protection and recorded as Indicator 9.3. Indicator 9.3—Family contact refers to the views of young people in OOH care aged 8–17 years about the frequency of contact with family members. Data from a survey of children in 2018 found that 72% of children reported satisfaction with one or more contact types (i.e., visiting, talking, or writing) with non-coresident families.

Information about health checks of children in OOH care could not be located when the author searched the AIHW website in February 2025. This indicates a need for policy to ensure that monitoring of standards occurs.

AIHW [24] discussed the circumstances of children described as “cross-over children” who became involved in the youth justice system due to an alleged offense, as well as being in the child protection system. Cross-over children were aged 10–17 years. The report found that, while most of the children referred to the child protection system due to abuse and neglect did not go on to offend, a large proportion of children who offended had a history of being notified for abuse or neglect. Young people who had experienced an out-of-home placement were more likely to be convicted of a crime than other young people in the general population. One report found that 74% of convicted young people aged 10–17 years had not offended before being placed in out-of-home care, and 61% of young people aged 10–17 who experienced residential care had not offended prior to placement, and they committed their first offense either during or after their first residential care placement. The report found that 28% of young people aged 10–17 years under youth justice supervision during 2020–21 had an interaction with the child protection system during 2020–21.

Further studies have examined the cohort of ‘crossover children’. Baidawi and Sheehan [25] expressed concern about the over-representation of children who had a child protection background and who were in the youth justice system. They conducted a detailed case file audit of 300 children who appeared before the Victorian Children’s Court in 2016–17. They found that young people who left the care of child protection were at least nine times more likely than other young people to offend and come under the supervision of youth justice services, and they spoke of a ‘care-to-custody pipeline’ between the child protection system and youth justice system.

The case file analysis analyzed the adverse childhood experiences of the young people involved. Their findings were as follows: 73% had been exposed to family violence; 12% had experienced the death of a sibling or other family member, including from overdose, suicide, and homicide; and 50% had a household family member with a severe mental illness. While exposure to parental substance abuse was reported, incident figures were not provided. These data show that crossover children in the sample had multiple needs.

An assessment was made of the number of adverse events crossover children had been exposed to prior to removal from parental care, using a list of ten adverse childhood events. The study found that crossover children were exposed to a mean of 5.4 adverse events, with 68% of children having been exposed to five or more adverse events. The study found that children did not receive intensive support services until their behavior attracted serious youth justice sanctions.

The Baidawi and Sheehan study found that child protection services had received notifications about most crossover children before the age of 10 years, emphasizing the potential for the child welfare system to make referrals for early intervention therapy and support for this cohort of vulnerable children.

The Baidawi and Sheehan study found that 73% of crossover children had misused drugs and/or alcohol, and there was evidence that 40% had used crystal methamphetamine, other amphetamines, heroin, or inhalants. The report recommended greater use of diversionary options for crossover children.

Malvaso and colleagues [26] were commissioned to review information about the overlap of young people who were registered with both child protection (CP) services and youth justice (YJ) services in South Australia in the period of 1991–1998. They reviewed data for 47,377 young people in the child protection system and 3058 young people in the youth justice system. The main findings of the Malvaso study were as follows: (a) 84% of young people supervised by Youth Justice had contact with the CP system; (b) 40% of the YJ group had substantiated maltreatment and 24% had spent time in out-of-home care; (c) crossover children had experienced many forms of substantiated maltreatment; and (d) 96.3% of crossover children had contact with CP prior to being under YJ supervision.

Similar conclusions that there are links between child maltreatment and later criminal convictions were reported in a study involving 5399 youth in Quebec [27].

In conclusion, statistics provided by the AIHW and other sources support a need to improve policy to improve the assessment of children in families where parents misuse substances by introducing objective assessment instruments to ensure that vulnerable children who are referred to child protection services are referred to early intervention therapy and practical supports.

#### 3.3.10. vi–Child Protection Legislation

Child protection legislation recognizes that two very different types of intervention are required for families where a child is vulnerable due to substance misuse and exposure to family violence: An approach providing early intervention therapy and support for families motivated to improve their parenting practices and a child protection approach that safeguards child from an unacceptable risk of harm by removing a child from parental care. The South Australian Government has stated that policies providing early intervention therapy are the priority before a child is removed from parental care, so that removal of children from parental care is a last resort.

The South Australian child protection legislation (Children and Young People (Safety) Act 2017-CYPS) permits departmental staff to apply to initially remove a child from parental care, for 6 months while an investigation is conducted, and then for two further periods of 12 months, each while further investigations continue. Legislation permitting prolonged periods of assessment can be viewed as facilitating inadequate assessment practices.

#### 3.3.11. Empirical Research About Child Protection Practices

A report of the AIHW titled ‘Child Protection 2017–2018’ found that (a) 56% of children across Australia who are subjected to notifications are removed from parental care while allegations are investigated and are then returned to parental care; (b) 82% of children who are removed from parental care remain in foster care for over a year while investigations are conducted; and (c) of the children removed from parental care, 9 per 1000 are aged 1–4 years and 8.4 per 1000 are aged 5–9 years.

The length of time taken by child welfare staff to assess risk to children while litigation-oriented investigations are conducted implies the agency does not use efficient methods to assess risk to children. Research summarized by Forslund et al. [28] showed that long separations of children from a parent whom the child has developed an attachment bond with increases the risk that a child will develop a mental disorder. This information indicates there is a need for policy improvement regarding the objectivity of the assessment of children in families where parents misuse substances.

When a child is placed in departmental care, the child’s contact with their parent is commonly restricted to supervised contact. Departmental reports to the court indicate that staff assess the quality of interactions by observing the parent and child, without the benefit of any structured assessment framework. This information indicates there is a need for ongoing research into whether assessment practices used in child welfare services are most suited to meeting the best interests of children, including the overall safety of children.

#### 3.3.12. vii–Health Practices

The South Australian Children and Young People (Safety) Act 2017 (CYPSA) include a section that requires health and education professionals to make a mandatory notification to the child protection department if they have a reasonable suspicion that a child is at risk of harm. The mandatory notification requirement passes information in one direction only, as the welfare department is not obliged to communicate information to health or education departments unless a Freedom of Information form is submitted and approved.

Further, the mandatory notification requirement does not provide any obligation to health professionals who make notifications to provide health services themselves or to refer a notified person to other treatment agencies. There is a need for policy adjustment to ensure that health professionals who make notifications that a child is at risk of harm also provide or refer families to early intervention therapy.

#### 3.3.13. Empirical Evidence About Health Practices

No research was identified about practices by health professionals when they made a notification to child welfare authorities about suspicions of harm to a child, including whether the health professionals themselves provided a therapy service to the child or family. There is a need for research into whether health services that make notifications to child welfare services make sufficient effort to provide vulnerable children and their families with early intervention therapies.

#### 3.3.14. viii–Law Enforcement Practices

Law enforcement agencies have a role to play when parents engage in practices that significantly increase the risk that a child might be exposed to family violence [14]. Processes in Australia involve three courts. Allegations can be made to the police and be referred to a Magistrate’s Court. Allegations made by separated parents can be made to a Family Law Court. Allegations that a child is not safe with a parent are referred to a Child Protection Court.

Under Australian legislation, the Commonwealth Family Law Court and State Child Protection Courts are viewed as civil courts where the standard of proof required to substantiate an allegation is on the ‘balance of probabilities’ rather than the higher ‘beyond reasonable doubt’ standard that applies in a criminal court.

Civil courts are authorized by legislation to issue a range of orders including restricting access between a parent and child; requiring a supervised changeover when care of a child is transferred from one parent to the other; requiring a parent to refrain from specific actions such as using a substance before and while a child is in their care; requiring a parent to participate in physical testing for substance use; or prohibiting contact between a parent and child.

#### 3.3.15. Criteria for Family Assessors

Commonwealth legislation sets the criteria required for the appointment of family assessors who provide reports about the functioning of families to family law courts. Australia follows the Daubert criteria for the admissibility of expert evidence that refer to (a) whether an assessment procedure can be tested; (b) whether the procedure has been subjected to peer review and publication; (c) the existence and maintenance of standard procedures for administering an assessment; and (d) an instrument’s known or potential rate of error in making predictions.

The Australian Supreme Court (Makita Australia Pty Ltd. v Sprowles, 2001, 52 NSWCA 305, Sydney, Australia) has recognized further criteria for the admissibility of evidence from an expert. The Makita criteria for admissibility of evidence can be summarized as follows: (a) there must be an established field of specialized knowledge; the expert must demonstrate their knowledge of the field based on specified training, study, or experience; an expert must establish their opinion is wholly or substantially based on their expert knowledge; a distinction is made between accepted facts and observations; and the scientific or other basis of conclusions must be established. It is the role of a court to decide the weight to give to admitted evidence.

#### 3.3.16. Evidence About Evaluation of Reports by Family Assessors

Lewis et al. [29] interviewed professionals who provide family reports to judicial services in Queensland Australia and found that reports provided by some child protection caseworkers were not highly regarded by courts. The report identified a need to improve the quality of expert reports by providing guidelines and standards for family assessment experts who work in child protection.

Carson et al. [30,31] interviewed 61 young people aged between 10 and 17 years whose parents had separated, along with 47 of their parents, about their experiences and needs. Most young people (62%) had some engagement with mental health professionals and services and found this helpful. The main concerns of children regarding their parents involved alcohol or substance abuse (21%), emotional abuse (64%), mental health issues (61%), and violent or dangerous behavior (32%). The young people stated the following opinions: 76% wanted parents to listen more to their views related to parenting arrangements; young people wanted their views to be taken seriously by family law professionals, including family consultants and independent children’s lawyers, particularly when they raised safety concerns; and young people asked to be kept informed about aspects of the legal process. The authors recommended greater use of children-inclusive approaches in court proceedings.

van IJzendoorn et al. [32] and van IJzendoorn, Steele, and Granqvist [33] emphasized the need for assessors who submit reports to family-oriented courts to ensure the evidence they provide meets both scientific and legal standards of evidence. There is a need for policy makers to review whether assessors who provide reports to family-oriented courts meet standards that are being identified in international research and whether assessment practices promote liaisons between health professionals and legal practitioners.

#### 3.3.17. Empirical Evaluation of Efficacy of Court Orders

Three studies were found that examined the efficacy of orders made by courts involving parents who misused substances.

De Bortoli et al. [34] analyzed files of 273 cases in the Children’s Court of Victoria, where children had and hadn’t been removed from parental care, to assess the level of parental compliance with court orders regarding substance use. They found that parental substance misuse was present in 51% of the sample, and poly-substance abuse was common. Parental substance abuse was associated with lower rates of parental compliance with court orders and with longer durations between notifications and final decisions by the court. The authors concluded that parental substance misuse and non-compliance with court orders were significant factors in delaying stability for a child through the granting of court orders that might involve child removal, indicating that prompt recognition of parental substance misuse and engagement with therapeutic services is required to protect children from delayed decision-making.

Carson et al. [31] noted that police work with Magistrate’s Courts that are established under State legislation, meaning that Family Law Courts that are established under Commonwealth legislation have limited capacity to enforce orders they make.

Isobe et al. [35] reviewed literature involving mothers who misused substances and found that mental health issues and family violence often co-occurred with substance misuse in this cohort. They considered that neither agency is enabled to manage the important issue of ensuring that the impacts of one parent’s ongoing violence are addressed sufficiently, and they drew attention to the need to strengthen collaboration between the three sectors involving substance misuse, mental health, and family violence.

One potential barrier to collaborative practice is that the agencies of child protection, substance misuse, mental health, and law enforcement are staffed by professionals who are trained in different disciplines, introducing a possibility that members of each discipline favor interventions that require involvement of their own discipline and prioritize the interests of their discipline over the best interests of children [36]. A further barrier is that professional codes of ethics and legislation promote the importance of maintaining the confidentiality of personal information, leading to professionals being reluctant to share information with people from different disciplines who are employed in different agencies.

Difficulties in achieving multi-disciplinary collaboration between services escalate when a client displays multiple issues and difficulties are heightened when an adult is designated as having multiple complex needs, as they meet the criteria to be eligible for services from several departments. Questions then arise about how to coordinate service delivery for people with multiple needs who are eligible for several services. The issues of confidentiality and collaboration are discussed below.

### 3.4. Empirical Evaluations of Child Welfare Practices

One policy is that children are removed from parental care as a last resort after early intervention measures have been provided to a family. AIHW data can be used to evaluate whether an adequate threshold is set between providing early intervention therapies for children in vulnerable families and the removal of children from parental care.

There is concern both about the length of time taken by welfare personnel to investigate allegations that a child is at risk of harm after a child has been removed from parental care, and about methods of investigation used. AIHW reports information submitted by States about indicators on its website under the title ‘child protection Australia.’ However, the length of time to complete assessments is not included as a national standard.

AIHW [22] reported that an agreement on a nationally consistent definition of out-of-home care was reached in 2019. AIHW reported that the number of children placed into out-of-home care and on third-party orders had risen consistently from 2015 to 2017.

AIHW [22] reported that in each year about 3% of all children aged 0–17 years were abused and were assisted by Australia’s child protection systems. The frequencies of each type of substantiated abuse of children in 2019–2020 were as follows: emotional abuse in 54% of notified cases, neglect in 22% of cases, physical abuse in 14% of cases, and sexual abuse in 9% of cases.

AIHW [24] found that some children who had experienced out-of-home care continued to experience vulnerabilities after discharge from State servives, and they were at a higher risk of experiencing poor outcomes in areas including employment, involvement in the criminal justice system, and housing.

The statistics provided by the AIHW support the case for further research about whether vulnerable children who are referred to child protection services receive adequate assessment using objective screening instruments to identify vulnerabilities that are likely to be responsive to early intervention therapy and practical support, including identifying suitable screening instruments.

#### Is Out-of-Home Care Over-Used?

Data published by the AIHW can be used to assess whether the policy principle of removing children from parental care is a last resort and whether there is an adequate balance between providing early intervention therapies for vulnerable children and their parents and removing children from parental care.

There is concern about the length of time taken by welfare personnel to investigate allegations that a child is at risk of harm after a child has been removed from parental care. While the Australian Government introduced National Standards for out-of-home care in 2011, the length of time to complete assessments is not included as a national standard.

There is scope for international research about the rate of removal of children from parental care, also called termination of parental rights [37]. There is also scope to review policies about criteria used by courts involving judgments that a family has received sufficient early intervention therapy, and that it is necessary to remove a child from long-term parental care to ensure the overall safety of the child.

### 3.5. C–Assess Severity of Substance Use

The studies reviewed above show that parents who misuse substances are a heterogeneous group who have differing needs and often multiple needs. The literature emphasizes the importance of assessing the levels of severity of substance misuse, as recommended interventions vary according to severity. For example, ICD-11 distinguishes three levels of severity of substance use: episodic harmful use; substance misuse; and substance dependence or addiction. The National Institute on Drug Abuse (NIDA) views substance misuse as unhealthy use that produces effects that produce long-term harm, and the term addiction describes the most severe end of the spectrum where a condition is a chronic and relapsing disorder.

The World Health Organization produced a 10-item AUDIT instrument that provides objective assessments of alcohol consumption, drinking behaviors, and alcohol-related problems. A threshold is set to indicate hazardous or harmful alcohol use. The World Health Organization also produced an instrument for use in primary health care to facilitate brief interventions or referrals [38].

Conway et al. [39] reviewed several instruments that had been proposed to assess severity and propensity to use substances. They favored the use of a Severity of Dependence Scale with five items that are self-rated on a 4-point Likert scale. Threshold scores are available for different substances. Ridenour et al. [40] proposed the use of a Transmissible Liability Index for young adults. McNeely et al. [41] recommended an instrument for use by primary care physicians that rates risk into three categories (low, moderate, and high) with the aim of improving universal screening of substance misuse, promoting a harm minimization approach, and minimizing child abuse in New York.

This review found that no instrument was used consistently in studies to assess the severity of substance use, and reviewers commented that the heterogeneity of samples made it difficult to draw clear conclusions about the efficacy of interventions.

This review proposes that two things are required to provide an adequate assessment of the severity of parental misuse of substances; first, a model is required that identifies risk factors to be assessed, and second, an instrument is required that provides objective assessments of the level of severity of each risk factor and the associated likelihood of harm to children. There is a need for ongoing research to clarify risk factors for children whose parents use substances. There is also a need for policy development to promote the use of assessment instruments that have been validated to provide objective assessments in families where parents misuse substances.

### 3.6. D-Risk Factors for Harm to Children

van IJzendoorn et al. [42] reviewed thousands of studies, including nearly 1.5 million participants, about risk factors for the abuse of children. The authors identified six important risk factors for the abuse of children that can be changed by the provision of appropriate support and that are relevant to parents who misuse substances. The identified risk factors are as follows: (a) intimate partner violence; (b) parental experience of maltreatment in their own childhood; (c) aggressive parental personality; (d) low socioeconomic status of the family; (e) parental emotionality as measured from high baseline autonomic nervous system activity; and (f) a dependent parental personality.

A review by the Australian Institute of Family Studies [30] identified further modifiable risk factors for child abuse, adding parental substance misuse; parental serious mental illness; parental exposure to multiple stressors; parents being isolated and lacking support; parents finding their child’s behavior difficult to manage; parents having difficulty managing their child’s special needs; and parents lacking a suitable parenting template after being raised themselves in an abusive environment, including in out-of-home care.

The research supports the proposition that a proportion of parents who misuse substances experience other risk factors and will be viewed as part of the cohort who have multiple complex issues.

To date, no Australian State Government appears to have used the research information cited above to formalize a policy about how to assess the risk of harm to children due to parental risk factors and to introduce the mandatory use of a standardized assessment instrument. There is a need for policy development about the assessment of risk to children arising from parental substance misuse that is evidence informed.

### 3.7. Empirical Evidence About Risks to Children

There is considerable research about harm to children whose parents misuse substances. In part, this research examines children who have been placed in out-of-home care.

Studies that examined the mental health of children in Australia who live in out-of-home care found that children removed from parental care and placed into out-of-home care have rates of mental disorders that are five times higher than the usual prevalence [43,44,45]. A meta-analysis by Dubois-Comtois et al. [46] analyzed 41 studies of children in out-of-home care and matched samples to clarify factors associated with children’s mental disorders. The study found an association between placement in foster care and a child’s psychopathology with an effect size of d = 0.19, with children in foster care having higher levels of psychopathology. The prevalence of psychopathology in children in foster care was like that of children who remained with their biological families, indicating that simply placing a child into traditional foster care is not a protective factor for children’s mental health.

A study by Vargas et al. [47] reported that children who observe domestic violence are at greater risk of repeating the cycle of violence when they become adults by entering abusive relationships or by becoming abusers themselves. They report that boys who see their mother being abused are ten times more likely to abuse their female partner when they are adults, and girls who grow up in a home where their father abuses their mother are likely to enter relationships with violent partners and are over six times more likely to be sexually abused compared to girls who grow up in a non-abusive home.

Kaspiew et al. [48] surveyed 10,002 separated parents in Australia and found that 17% of fathers and 26% of mothers reported experiencing physical hurt from their partner. Of the parents who reported experiencing physical violence before separation, 72% of mothers and 63% of fathers reported that their children had witnessed the violence. Similar figures were reported by Kaspiew et al. [49].

A systematic analysis of research about the impacts on children of having been exposed to a caregiver who misused substances was provided by Staton-Tindall et al. [50]. Their review found that researchers had used a wide variety of measures, making it difficult to compare studies. The review found that having a substance-using caregiver was associated with a higher rate of referrals and re-referrals to child welfare services and a higher rate of substantiated allegations of abuse. The reviewers found that children in households where parents misused substances were exposed to violence, chaotic lifestyles, and other high-risk situations, incurring the risk of childhood trauma. There was a higher level of child maltreatment by substance-misusing caregivers, with physical abuse and neglect being the most reported types of child maltreatment associated with caregiver substance misuse.

Caregiver substance use was found by Staton-Tindall et al. to be consistently related to high and clinically significant levels of mental health disturbances in children. The reviewers found that researchers had called for increased provision of evidence-based interventions for families where parents misused substances. However, the reviewers found that few researchers recommended children be referred to their own mental health therapist. The reviewers predicted that advances will be linked to improvements in screening instruments that measure key constructs, where instruments move beyond dichotomous measures. The reviewers noted that most studies report that agencies support either adults or children, with few studies reporting support for families, and this impacts reunification when children have been removed from parental care for assessment. The review called for the provision of therapies that are evidence-based, family-oriented, and trauma-specific to address co-occurring traumatic stress conditions such as parental PTSD, acute stress disorder, and a range of substance use disorders.

AIHW [5] reported that children who had been exposed to family violence displayed disadvantages on several measures including diminished educational attainment, reduced social participation in early adulthood, physical and psychological disorders, suicidal ideation, behavioral difficulties, homelessness, and future victimization and/or violent offending.

Orr et al. [51] analyzed the health and police records of 16,356 children who had been exposed to family violence in Western Australia between 1987 and 2010 to explore connections between exposure to family violence in childhood and contact with mental health services, compared to children with no involvement in family violence. The study found that children who had been exposed to family violence were almost five times more likely to have used a mental health service by the age of 18, with an increased risk of having been diagnosed with 8 of the 10 mental health disorder diagnoses examined in the study, including double the likelihood of having a substance use disorder.

In summary, research indicates that children exposed to comorbid parental substance misuse, parental mental illness, and family violence are at increased risk of developing their own mental disorders and forming adverse relationships. There is scope for international research about criteria used in various countries when determining that children are at risk of an unacceptable level of harm and need to be removed from care of their parents.

### 3.8. Cumulative Risk of Harm

One hypothesis proposes that there is an association between the number of risk factors that a family displays and the risk of cumulative harm to children. This is called a ‘cumulative risk hypothesis’.

Meyer et al. [52] conducted a case file analysis of 60 families who had been referred to a child protection court in Sydney, of whom half of the children were placed in foster care and half remained in parental care. Both groups of parents had alcohol and drug abuse issues. Parents whose right to care for their child was terminated had a higher number of risk factors including mental health problems, and they were more likely to have experienced incarceration.

Raviv et al. [53] studied a sample of 252 maltreated youths aged 9–11 years who were placed in out-of-home care in the USA to examine the cumulative risk hypothesis that a youth’s exposure to a higher number of risk factors was associated with an increased likelihood that the youth would develop mental health symptomatology. They identified seven risk factors as indicators that differentiated youth who did and did not score in the clinical range regarding mental health symptoms, confirming the cumulative risk hypothesis and proposing a potential threshold.

Raman and Sahu [54] reviewed the community health records of 57 children in foster care in Sydney to identify predictors that children were likely to be placed into foster care. They found that, compared to a control group, parental risk factors for a child to be placed into foster care included parental substance use (65%), domestic violence (57%), and parental mental health disorder (33%).

Solomon et al. [55] further examined the relationship between cumulative risk, parental recidivism, and provision of therapy for parents and children. Their study confirmed a hypothesis that providing therapy for parents that reduced the number of risk factors also reduced parental recidivism. The finding indicates that it may be viable for a scale that provides quantified scores to set a threshold that identifies families who warrant early intervention therapy, as well as set a threshold where risk is excessive.

As discussed above, Baidawi and Sheehan [25] conducted a case analysis of young people who had been involved with both child protection and youth justice services in Victoria. An assessment of the number of adverse events children had been exposed to prior to removal from parental care found that removed children had been exposed to a mean of 5.4 adverse events.

Vial et al. [56] proposed a system to categorize the levels of risk to children based on the number of risk factors that are substantiated in a family. They labeled risk categories as low, medium, and high. Families who present medium risk might be labeled as vulnerable families who are offered access to therapy services that aim to improve their inadequate parenting practices. Families assessed as presenting with high risk due to the number and severity of substantiated risk factors might be labeled as high-risk and have their child removed from their care for protective purposes while the parents are given opportunities to engage in rehabilitation.

Studies show that parental risk factors can cluster. One cluster of parental risk factors that increase the risk of harm to children involves four factors of parental substance misuse, parental mental health conditions, difficulties in managing children’s behavior, and family violence. This review proposes that to facilitate the delivery of effective services for vulnerable families, it is essential to adopt a model of risk that is relevant to vulnerable families and includes the parental risk factors that have been found to cluster.

### 3.9. Assessment Instruments

Several themes have been identified in the sections above including that certain parental risk factors tend to cluster, increasing risk factors that present a risk of cumulative harm to children, and there is a need for a valid screening instrument to distinguish between families where the provision of preventive therapy is appropriate and families where children need to be removed from the long-term care of their parents to protect the child from unacceptable harm.

Cafcass published a SCODA tool to assess the risk arising from parental drug use [57]. No research was identified in this review about the ability of SCODA to distinguish families who require early intervention therapy from families where children need to be removed from parental care to protect the child.

This review did not identify any instrument that has been established for the purpose of assessing the level of risk to children arising from parental misuse of substances. As noted above, the absence of an agreed objective instrument to assess risk to children contributes to some children being removed from the care of their parents for prolonged periods while informal assessments are conducted for litigation purposes.

Tustin and Whitcombe-Dobbs [58] provided a Parenting Capacity Instrument (PCI) that identifies risk factors for the maltreatment of children and appears relevant as a model for assessing the risk associated with parental substance misuse. The PCI includes a domain of major parental risk factors, with 13 items that identify modifiable parental risk factors and might serve as a screen. It is proposed that scores on the PCI have the potential to distinguish between parents who use adequate parenting practices; parents whose practices are not adequate to meet the needs of their child and who require distinctive types of therapy; and parents whose practices are abusive, where children need to be removed from parental care. However, the scoring system used in the PCI might not be adequate when applied to parents who misuse substances, and improvements might be made by further research.

This paper proposes that for a government to implement a policy of supporting families where children are vulnerable to harm from parental misuse of substances, it is necessary to adopt a model of risk that identifies specific risk factors and needs of vulnerable families and to identify an assessment instrument that provides objective assessments to ensure that each family is referred to appropriate services.

This paper proposes the Parenting Capacity Instrument and SCODA be considered as providing a model for assessing risk to children whose parents misuse substances. The topic of identifying an assessment instrument that is relevant to families where children are vulnerable due to parental misuse of substances requires ongoing research.

### 3.10. E-Scale of Problem

Administrative information is provided in Australia about the number of children placed into out-of-home care and related information.

The South Australian Department of Child Protection provides regular statistical reports on its website. Statistics about notifications for the year 2022–2023 show that 114,299 contacts were made to its call center; there were 92,951 reported notifications; and 39,515 notifications (43% of notifications) were screened in for investigation while the remainder were screened out for further action.

The Commonwealth Government, through its Australian Institute of Health and Welfare (AIHW) takes responsibility for monitoring the administration of welfare services that are provided by state governments. The AIHW reports statistics annually on topics including the use of out-of-home care where a court order is made for a child to be removed from the custody of their parents and placed in the custody of the child welfare department. Data reported over the years are summarized.

The AIHW [24] reported that the rate of notifications of children in Australia was 51 per 1000 in 2022–2023, indicating that 1 child in 32 were notified each year. Ten percent of notifications were made by health or medical personnel. About 33% of notified families were referred to another service by child welfare services. The AIHW [24] reported that the rate of substantiated notifications was 8 per 1000 for children aged under 18, where substantiation means that an investigation concluded the child had been maltreated or was at risk of being maltreated due to the establishment of risk factors.

### 3.11. F–Efficacy of Therapies

Authorities recommend that treatment of substance abuse be differentiated according to the severity of substance abuse by distinguishing between addiction and occasional use. Treatments for addiction and early intervention therapies are reviewed below.

#### 3.11.1. Treatment of Addiction

Reviewers have commented on the effectiveness of treatments when people are addicted to a substance. Laudet [59] noted that many treatments for addiction have been short-term and the benefits that are produced do not endure. Dennis and Scott [60,61] reviewed longitudinal studies of people who required inpatient treatment for heavy substance use and found that many users required three or four episodes of treatment before they achieved abstinence. They reported that 64% of people admitted to treatment programs in the USA were re-entering treatment. They identified four phases that users engaged in, which they called recovery/relapse/re-enter treatment/incarceration, and they found that users transitioned between each phase about every 90 days.

McLennan and colleagues [62] proposed that the most severe level of substance use disorder meets the criteria for being a chronic disease and needs to be managed using a proactive and ongoing chronic care model of intervention.

Simoneau et al. [63] reviewed 16 treatment studies and concluded that treatment of addiction requires ongoing long-term intervention rather than episodes of short-term treatment, confirming that addiction is a chronic disorder. Laudet [59] and McKay [64] identified requirements for effective interventions for addiction to be a person’s motivation to abstain, coping skills to manage stressors, a reliable source of emotional support, and attendance at a peer support group. A Delphi study of quality indicators for continuing care of people with addictions provided by Bekkering et al. [65] made 69 recommendations.

A meta-analysis of general psychosocial interventions for substance abuse disorders by Dutra et al. [66] found that one-third of participants who were addicted dropped out of therapy before treatment was completed.

Lopez et al. [67] reviewed 50 rigorous studies that examined group treatments for people diagnosed with drug use disorders involving cocaine, methamphetamine, marijuana, opioids, and mixed substances, with co-occurring psychiatric conditions. The review found that the efficacy of interventions varied with substances. They found that combined group cognitive behavioral therapy plus pharmacotherapy was more effective in decreasing opioid use than pharmacotherapy alone. Relapse prevention support groups, motivational interviewing, and social support groups were all found to be effective in reducing marijuana use.

#### 3.11.2. Goals of Treatment

Therapists can set different outcomes they aim to achieve from therapy. Kepple [68] surveyed families involving 5501 children in the USA who were identified as being at high risk of maltreatment and assessed levels of parental drinking in the previous year using four categories that included non-use/moderate drinking/risky use/substance abuse disorder (SUD). The survey found that, compared to non-use, each user category had a heightened incidence of physical abuse of children being 148% higher for moderate drinking, 386% higher for risky use, and 562% higher for SUD. However, the study found that the frequency of physical abuse by parents who had a previous diagnosis of SUD and who had reduced use was not significantly different from that of parents who reported non-use, indicating that changes are possible.

#### 3.11.3. Cannabis Use

One psychoactive substance that causes concern is cannabis. The Australian Department of Health and Aged Care provides advice about safe levels of consumption of alcohol based on a standard drinks approach that provides guidelines for the public and therapists and that can be adapted for specific population groups including parents. However, this review did not find guidelines about the safe consumption of cannabis, leaving judgments about goals to be set in therapy to the personal opinions of practitioners.

Some studies have focused on parental use of cannabis. Donoghue [69] interviewed 43 parents in Western Australia who used cannabis about strategies they used to minimize cannabis-related harm to themselves and their children. Most parents did not believe that their children had been adversely affected by their use of cannabis. Strategies used by parents to minimize harm involve the following: dosage control, managing substance dependency, awareness of acute risks, addressing long-term harm, and monitoring harm to children.

Madras, Han, and Wilson [70] noted that adults of childbearing and child-rearing ages were in cohorts with higher levels of cannabis consumption. They surveyed 24,900 cohabitating parent–child pairs in the USA about marijuana use by parents in the past year. They found that 7.6% of mothers and 9.6% of fathers of adolescents have used marijuana in the past year. They found that adolescents whose parents had used marijuana 52 or more days in the past year had higher relative risk rates of consuming marihuana than adolescents whose parents had not used marijuana.

Wesemann, Wilson, and Riley [71] surveyed 266 parents of children aged 1–5 to 5 years in three states of the USA about their use of cannabis in the past 6 months, parenting practices, exposure to adverse childhood events (ACEs), and children’s emotional/behavioral adjustment. Hierarchical regression analyses were conducted. The study found that 13% of parents reported cannabis use. Parents who endorsed the use of cannabis reported significantly more negative parenting, ACEs, anxiety, depression, and child emotional/behavioral problems. Adjusting for the effects of other risk factors, cannabis use significantly predicted more negative parenting practices.

Kokotovic, Psunder, and Kirbis [72] interviewed 839 secondary students in Slovenia aged 14–21 years about their use of cannabis. The study found the strongest predictor of student use of cannabis was parental use of cannabis.

Freisthler, Thurston, and Price Wolf [73] obtained self-report surveys from 77 parents who used cannabis over a 14-day period. They reported that different patterns were followed by groups of parents, and they concluded there were complicated relationships between cannabis use and parenting in their sample. Some parents provided information about the use of harm-reduction practices to support positive parenting.

In summary, several studies associated parental use of cannabis with negative outcomes for children.

#### 3.11.4. Early Intervention Therapies

Loxley et al. [16] were commissioned by the Australian Government to review evidence about the prevention of substance abuse. They supported universal educational programs for the whole community and targeted interventions for cohorts at higher risk. They reported that effective programs for at-risk cohorts provide early interventions that are focused on addressing specific risk factors in individuals in the cohort. The outcome measure proposed by Loxley et al. was alcohol use in the last 30 days. Family risk factors were identified as parental attitudes favorable to drug use, poor family management, poor discipline, family conflict, family history of antisocial behavior, and parental attitudes favorable to antisocial behavior. Family protective factors were attachment, opportunity for pro-social involvement, and rewards for pro-social involvement.

The section below reviews the literature about evidence of the efficacy of therapies used with parents who misuse substances, including in families where parents have complex presentations. Interventions are presented using a model proposed to assess the severity of the risk to children arising from different parental risk factors.

#### 3.11.5. Model to Assess Parental Risk to Children

This review proposes that to facilitate the ethical delivery of interventions to families that are appropriate to meet the individual needs of each family, it is necessary first to identify a model that identifies parental risk factors for children. Establishing a model of risk will facilitate the development of a screening instrument that provides objective assessments of risk to each child and that guides referrals to services that are relevant to each family’s individual needs.

Velleman and Templeton [74] provided information about protective factors and risk factors for children in families where parents consume substances. An effective screening tool will combine both risk factors and protective factors on one scale.

A proposed assessment model is summarized in Table 1. Table 1 describes three components of the model, which include labels for each risk level, eligible family, and types of intervention.

Seven levels of risk are proposed in the model: (a) no risk, so families access universal services available to all members of a population; (b) low risk, where families are eligible for identified services; (c) moderate risk from a diagnosed disorder, where a family is eligible for focused therapies provided by a clinician; (d) moderate risk of cumulative harm, where families are eligible for focused services; (e) moderate risk and reportable therapy, where a person receives focused services that are set by court order and by assessment; (f) moderate risk due to multiple complex conditions, where a family receives intensive support services supervised by a case manager and provided by a multi-disciplinary service; and (g) unacceptable risk, where a child is removed from parental care.

This paper uses the following format to describe administrative interventions. Interventions are distinguished according to the stage of development of a condition: *universal* prevention aims to stop a health condition from developing; *early intervention* services that are indicated aim to teach self-help skills when a health condition is emerging; and *treatment* is used when a health disorder has been diagnosed. *Targeted* health interventions are directed at specific groups in a population. *Indicated* and *focused* interventions are used with people who have begun to engage in hazardous activities with the aim of preventing heavy or chronic use of a substance and the aim of developing protective factors and reducing risk factors. Providers of indicated and focused interventions set priorities between goals when a person exhibits multiple concerning behaviors and engages in several hazardous activities.

Type of intervention refers to administrative arrangements rather than to clinical interventions, as similar clinical interventions are provided as part of different administrative interventions. The proposed model integrates four administrative interventions: *universal* interventions that are available to the public; interventions that are *indicated* for certain cohorts in a population where access might be restricted; interventions that are *focused* and restricted to designated groups; and interventions that are *legally mandated*.

### 3.12. Types of Therapy Intervention

The review identified a range of early intervention therapies and supports that have been provided for families where a child is at risk of harm due to parental substance misuse. Therapy interventions and supports are named in the third column of Table 1 and are described below, together with citations of the literature. It is noted that interventions require differing numbers of sessions and differing qualifications of providers, so there are cost aspects to selecting interventions that are not addressed in this article.

The following interventions are discussed: (i) psychoeducation; (ii) social support groups; (iii) therapies for substance use; (iv) joint parent–child therapy; (v) dual-focus therapies; (vi) trans-diagnostic therapy for co-morbidities; (vii) clinician-led therapy; (viii) reportable therapy; (ix) combined psychotherapy and pharmacotherapy; (x) integrated multi-disciplinary clinics; (xi) continuing case management; (xii) intensive family supports; and (xiii) co-located workers.

#### 3.12.1. i–Psychoeducation

Psychoeducation about the impacts of using substances is a commonly used intervention as it can be delivered to a group in a few sessions [75]. Psychoeducation for parents includes information about the impacts of the misuse of substances on children. 

Magill et al. [76] reviewed the literature that provides guidelines on delivering psychoeducation about substance use. They identified nine principles and twenty-one practices to encourage clients to engage in discussion and disclosure of their personal issues, rather than to adopt a didactic approach that seeks compliance. Effective groups encourage participants to discuss very personal issues in individual therapy that helps the person change strongly held beliefs and attitudes and to manage strong emotions.

The aim of a psychoeducational group is to improve an individual’s awareness of the behavioral, medical, and psychological consequences of substance abuse. Psychoeducational groups provide information that is generally relevant to people with a common need, including developing an understanding of the process of recovery. Groups encourage members to exchange information that is directly relevant to each other’s lives, including identifying community resources that can assist clients. Psychoeducation is considered to be more effective when a person is in a pre-contemplation stage of changing their activities, and when all members of a group are in a similar stage of change.

A social support peer group has the additional aim of motivating a person to disclose personal information when they feel safe, make commitments to other members to change, and reduce anxieties about participating in personal therapy. Rehabilitation groups that provide peer support assist in preventing relapses by providing group support in finding ways to manage current challenges.

Thylstrup et al. [77] reported a study where psychoeducation about substance misuse was provided to a treatment group but not to a control group. They found that people allocated to treatment groups attended a median of two of six scheduled psychoeducation sessions. Attendees showed reduced substance use at a 9-month follow-up. Aggression declined in participants of both groups.

Lyman et al. [78] conducted a meta-analysis of 30 studies involving randomized controlled trials of consumer psychoeducation about the impacts of psychoeducation involving clients with co-morbid severe mental illness and substance misuse. They found that psychoeducation improved adherence to recommended medication regimes, reduced relapse and hospitalization rates, and was a powerful intervention. The involvement of supportive family members in psychoeducation reduced burdens on family members who became involved.

Koc et al. [79] used a semi-structured questionnaire to assess change and found that an eight-session psychoeducation program was effective in improving parents’ knowledge about the adverse impacts of substance use on their children and in reducing parental symptoms associated with their use of psychoactive substances.

Reif et al. [74] concluded that studies had not yet tested the key mechanisms of the psychoeducation intervention.

#### 3.12.2. ii–Social Support Groups

The literature about a social support program that operates in Australia called Mirror Family was reported by Tsantefski et al. [80,81]. The Mirror Family is designed to help mothers affected by alcohol and other drug use. The aim of the program is to introduce a vulnerable family to a social group that will function as an extended family for the life of the child.

The service provides regular home visits from a qualified worker, initially on a weekly basis for two hours, and then reducing in frequency to fortnightly and then monthly. The duration of support for individuals ranged from 7 to 22 months, with a median of 7.5 months and a mean of 14 months. The focus of support is on helping the mother to meet caregiving responsibilities and to ameliorate the negative effects of parental substance use on the family. The program includes helping parents to reconnect with their community and with family. One emphasis of the program is to buffer children from the impacts of parental problems by creating an environment with an extended family for life. The Mirror Family program provides three main types of support. One type of support offers respite care for parents. A second support offers a diversity of supporting roles, including baby-sitting, attending family celebrations, accompanying the child to sporting and other events, and mentoring and advocacy. A third type of support offers direct care.

Tsantefski, Humphreys, and Jackson [81] recognized that babies and infants are extremely vulnerable during the first year of life, and they provided a report about a program that followed 20 substance-dependent mothers from the perinatal period until infants were aged 12 months, using a risk assessment and management approach. Their study identified a sub-group of infants who were at an increased risk of harm during this very vulnerable period, and they argued for a differential response to the management of risk for this cohort of infants by addressing family-related risk factors when mothers were most open to help. Data was gathered using semi-structured interviews.

#### 3.12.3. iii–Therapies for Substance Use

Several therapies that are derived from different theoretical perspectives have been shown to be effective in treating people with a substance use disorder. Studies vary in their stated goals. Some studies state a treatment goal for sustained abstinence for a defined time such as a year. Other studies involving alcohol aimed to reduce consumption to a recommended number of standard drinks over a 30-day period. Studies vary in the follow-up periods reported.

A study by Forray et al. [82] involving 152 pregnant women identified the likelihood of relapse as a measure of the severity of substance misuse. They proposed that the number of weeks a person remains abstinent from a substance can be used as a measure of the severity of dependence on the substance. They reported the following mean numbers of weeks before relapse by mothers who aimed to abstain postpartum: 16 weeks for cigarettes; 18 weeks for alcohol; 20 weeks for cannabis; and 41 weeks for cocaine. Forray et al. reported that the USA National Institute of Drug Abuse estimates that overall relapse rates for substance use disorders are between 40% and 60% and consider that these relapse rates are comparable to relapse rates for other chronic medical conditions such as hypertension, diabetes, and asthma. Forray et al. proposed that addiction to a substance be viewed as a chronic health disorder.

The Substance Abuse and Mental Health Services Administration’s data compiled for the decade 1994–2004 provide information about relapse rates associated with specific substances. Reported relapse rates were as follows: 78.2% for heroin use; 68.4% for severe alcohol use disorders; 61.9% for cocaine use disorder; and 52.2% for methamphetamine use disorder.

Davis et al. [83] summarized 10 randomized controlled trials that used psychological interventions for cannabis use disorders. They found that effective interventions achieved a combined efficacy of Hedges’ g = 0.44 and these were behavioral treatments involving contingency management, relapse prevention, motivational interviewing, and cognitive behavioral therapy (CBT).

Cooper et al. [84] reviewed 33 random controlled trials of psychological interventions for cannabis use. They found that CBT interventions reduced cannabis use and severity of dependence in 4–14 sessions, with benefits continuing during a follow-up period of 9 months. Evidence-based therapies for cannabis use identified in the review by Cooper et al. were cognitive behavioral therapy, motivational interviewing and motivational enhancement therapy, supportive-expressive dynamic psychotherapy, and social support groups. Cooper et al. reported that studies were heterogeneous and covered a wide range of interventions, comparators, populations, and outcomes.

Studies reviewed by Cooper that involved brief motivational interviewing with only two sessions produced mixed results. Combining CBT with the provision of vouchers for abstinence showed some efficacy. When cannabis users had comorbid mental health diagnoses of bipolar disorder, major depression, or psychosis, CBT did not add to the efficacy of standard treatment for the mental health disorder.

Gates et al. [85] reported a meta-analysis of 23 studies involving cannabis use where participants received a mean of seven sessions of treatment. The review found that 70% of participants completed treatment with outcomes of fewer days of cannabis use and higher rates of abstinence compared to a control group, provided participants engaged in four sessions of treatment. The review found that samples were heterogeneous in terms of participants involved, intensity and duration of treatment provided, and outcomes sought.

DiClemente et al. [86] reviewed the efficacy of motivational interviewing for addiction to various substances, and concluded there was evidence of efficacy for alcohol, marihuana, and tobacco use. DiClemente et al. reported that results about effectiveness were mixed due to limited assessments of fidelity, limited analyses of subpopulations, and differences in dose, outcomes, and protocol specifications.

Hogue et al. [87] cited meta-analyses that compared the efficacy of therapies that involved family members in the treatment of substance users and reported a meta-analysis of family-involved treatments by Ariss and Fairburn [88] that condensed data from 2115 adolescents and adults across 16 independent trials who calculated a small effect size that endured up to 12–18 months post-treatment and translated to a 5.7% reduction in frequency of using substances.

A meta-analysis of the efficacy of psychological therapies for substance abuse disorders involving alcohol, cannabis, stimulants, opioids, and benzodiazepines was reported by Dellazizzo et al. [89]. They found evidence of efficacy for interventions of brief intervention, cognitive behavioral therapy, contingency management, voucher-based reinforcement therapy, motivational interviewing, motivational enhancement therapy, involving significant other people in therapy, and cue-exposure therapy. Effects were small-to-moderate for motivational approaches and cognitive behavioral therapy with cannabis use disorder. Small effects were observed for contingency management as well as cognitive behavioral therapy for amphetamine-type disorder. Small effects were found for contingency management for cocaine use disorder. Moderate effects were found for contingency management and voucher-based reinforcement intervention for opioid use disorder.

Nadkarni et al. [90] reviewed the efficacy of different psychological therapies for alcohol use disorders. They found demonstrated efficacy for eight interventions: behavioral couples therapy, cognitive behavioral therapy combined with motivational interviewing, brief interventions, contingency management, psychotherapy plus brief interventions, 12-step programs, family therapy/family-involved treatment, and community reinforcement. Common components in effective programs were individual assessment, personalized feedback, motivational interviewing, goal setting, setting and review of homework, promoting problem-solving skills, and relapse prevention/management.

SAMHSA [91] reported that a 2020 National Survey on Drug Use and Health found that over 22% of American adults reported illicit drug use in 2020 and that 6.8% of adults meet the criteria for a drug-induced disorder. They conducted a meta-analysis to identify interventions that are most efficacious for adolescent drug users. Their analysis concluded that interventions involving parents produced small-to-medium positive relations with youth substance use and psychological problems.

Klimas et al. [92] reviewed four studies where psychological interventions were used with people who consumed both alcohol and illicit drugs. Interventions studied were CBT coping skills training, a 12-step program, motivational interviewing, and brief intervention. Alcohol use was measured using an AUDIT instrument. Klimas et al. compared the efficacy of interventions for specific substances and aimed to assess the relative efficacy of interventions rather than efficacy in meeting stated treatment criteria. The review found that evidence was of low quality, and it was not possible to identify therapies that were superior. Therapies where there was evidence of efficacy for concurrent alcohol and illicit substance use were cognitive behavioral coping skills and motivational interviewing.

Haber et al. [93] provided guidelines for general medical practitioners regarding the assessment and treatment of substance misusers. They recommended using the Alcohol Use Disorders Identification Test (AUDIT) to screen for alcohol use that assesses the quantity and frequency of consumption and the impact of consumption on physical health, mental health, and social functioning. The AUDIT provides information to develop a treatment plan and a relapse prevention plan. The authors identified pharmacotherapy to maintain abstinence, individual therapy, and peer support involvement as effective in reducing alcohol use. Engaging family and culture-specific agencies were recommended. Harm minimization by reducing consumption was considered an alternative goal to abstinence. Neuropsychological assessment was recommended if significant cognitive impairment was suspected.

In conclusion, the literature review identifies a range of specific therapies for substance misuse. Reviews report that samples in studies are heterogeneous, making it difficult to draw firm conclusions about sub-groups where interventions are more and less effective. It is proposed that individual parents who misuse substances can be referred to a skilled clinician to treat the condition, leaving the selection of suitable treatment modalities to the judgment of an evidence-oriented clinician.

#### 3.12.4. iv–Joint Parent–Child Therapy

Some programs focus on improving parenting skills in families where parents misuse substances and children are at risk of physical violence and provide joint parent–child therapy [5].

Shulman et al. [94] provided an outreach program for children whose parents were in a treatment program for substance abuse, where children were at risk of developmental delay. Of the children referred for psychological testing, 50% scored in the borderline range of intellectual functioning, 68% demonstrated a variety of speech and/or language impairments, 16% were diagnosed with emotional or behavioral disorders, and 83% were found to have medical or nutritional disorders or both. The program provided individual assessments of children’s skills and children were referred to individualized therapy, resulting in 72% of children receiving services.

Schuler et al. [95,96] provided an in-home intervention for post-partum mothers who used drugs that provided weekly visits for 6 months, followed by fortnightly visits for 6 to 18 months. The aim of the intervention was to educate mothers about child development. The program did not aim for mothers to be abstinent. The study found that ongoing maternal drug use and poor parenting attitudes were associated with less optimal maternal behaviors during mother–child interactions.

Dawe and Harnett [21] reported on a Parents under Pressure (PUP) program for 64 mothers who received methadone and whose children were aged 2 to 8 years. PUP is an individualized intensive home visiting program with 12 modules that aims to improve parenting skills by addressing three main issues: psychological functioning of individuals in the family, including teaching parents to regulate their emotions; parent–child relationships; and addressing social/contextual factors. The program was implemented by professional clinicians. An evaluation of the program found that, compared to a control group, parents who completed the program had reduced child abuse potential, experienced less stress from the parenting role, and showed improved parental emotional regulation, and their children’s behavior improved.

Coates [36] described a treatment program provided by a multi-disciplinary team of clinicians called the Keep Them Safe-Whole Family Team (KTS-WFT), which provided therapy for children whose parents exhibited drug and alcohol and/or mental health issues. Referrals were accepted from child protection services. The study focused on the need to improve collaboration and information sharing, overcome silo ways of thinking, manage risk together in more consistent ways, and develop consistent processes and expectations.

Ritzi et al. [97] reported that no specific intervention for working with parents who misused substances had previously been identified, and they proposed a model of intervention for families where parents misused substances and placed children at risk of violence.

In conclusion, a range of interventions have been reported that provide joint therapy for children and their parents. Families who have received joint parent–child therapy are heterogeneous, and outcomes are often not specified apart from citing improvements compared to a control group. There is insufficient evidence to support any specific programs over alternatives.

#### 3.12.5. v–Dual-Focus Therapy

Some studies have been designed to treat people with dual diagnoses of parental substance misuse and other health conditions. Hides et al. [98] reviewed evidence from seven random controlled trials about the efficacy of four psychological interventions for people who had comorbid depression and substance use disorders, with the aim of identifying interventions that are more effective. The interventions studied were the following: Integrated Cognitive Behavioral Therapy (ICBT), Twelve Step Facilitation (TSF), Interpersonal Psychotherapy for Depression (IPT-D), and Brief Supportive Therapy (BST). One of their meta-analyses compared ICBT with TSF and found that while the TSF group had lower depression scores at post-treatment follow-up after 12 months, the difference was not statistically significant. No significant differences were found between groups in proportion of days abstinent, although the ICBT group had a greater proportion of days abstinent at the 12-month follow-up.

A second meta-analysis by Hides et al. compared IPT-D with BST. IPT-D produced immediate significantly lower depression symptoms, but the benefit was not sustained at a 3-month follow-up. Substance use was not reduced by either intervention as interventions did not specifically address substance misuse. The review by Hides and colleagues found some evidence of efficacy for integrated cognitive behavioral therapy, 12-step facilitation, and interpersonal psychotherapy for depression with clients who had comorbid depression and substance abuse disorders.

Vulanovic et al. [99] examined efficacy of therapy for people with comorbid depression and substance use disorders and concluded that more work is needed using a transdiagnostic therapy approach to evaluate efficacy of therapy with this comorbid condition.

Cleary and colleagues [100,101] reviewed meta-analyses that assessed impacts of substance misuse by people with a severe mental illness and found that even low levels of substance misuse could have detrimental effects on people with a severe mental illness. They found no compelling evidence to support any psychosocial treatment over other treatments to reduce substance use or improve mental state for people with serious mental illnesses. 

#### 3.12.6. vi–Transdiagnostic Therapy for Co-Morbidities

Many studies find that the cognitive behavioral therapy (CBT) approach is effective in treating substance use disorders alone. The CBT approach encourages clients to review their thinking and emotional reactions to challenges and to find new coping strategies apart from the use of substances. The CBT approach is known for following a scientist–practitioner model, where there is two-way communication between clinicians and scientific researchers, clinicians aim to use therapy interventions that are soundly based on principles established in research, and scientists consult with clinicians about topics that require further research.

Commentators note that the CBT approach produces regular advances in knowledge and that clinical practices derived from the CBT approach progress through phases. In the first phase, clinicians focused on treating single disorders. In subsequent phases, clinicians accept referrals for clients who have been diagnosed with a set of disorders that occur in clusters, and clinicians use a trans-diagnostic approach where they select treatment methods according to each client’s presenting issues and needs, rather than being restricted to set procedures that are associated with a single diagnosis [102]. Trans-diagnostic therapies focus on mechanisms that contribute to the development and maintenance of psychopathology. Clinicians who use a trans-diagnostic approach are willing to provide interventions that are derived from different theoretical models if an intervention is clear and is shown to be effective. Some therapists focus on interactions between people or on interpersonal dynamics [103], while another trans-diagnostic approach for substance misuse examines people’s strongly held beliefs using schema therapy [104].

The trans-diagnostic therapy approach is beginning to be applied to the treatment of substance misuse. Sudhir [105] discussed one form of trans-diagnostic therapy using relapse prevention strategies based on teaching clients to recognize and manage their signs of cravings. Coping skills used in managing cravings included decreasing the valence of addictive behaviors, teaching coping skills to manage cravings, managing high emotional arousal and negative mood states, assertiveness skills to manage social pressures, family psychoeducation, cognitive strategies to enhance self-efficacy beliefs, and modification of expectations about outcomes of addictive behaviors. Skills used in trans-diagnostic therapies for substance disorders include building distress tolerance skills using mindfulness practices.

Kim and Hodgins [106] described a trans-diagnostic model for treating addictions that addressed the client’s low motivation, deficits in self-control, deficits in social support, and compulsive tendencies.

Narayanan and Naaz [107] identified a range of trans-diagnostic therapies that are used to treat substance use disorders including Acceptance and Commitment Therapy, Dialectical Behavioral Therapy, Metacognitive Therapy, and Mindfulness-Based Relapse Prevention. They report growing evidence for the efficacy of trans-diagnostic therapies in treating substance misuse disorders.

Neger and Prinz [108] reviewed 21 studies that provided dual treatment of substance misuse and parenting education and reported that the dual-treatment program generally produced both a reduction in parental substance use and improvements in parenting practices.

Moreland and Mcrae-Clark [109] reviewed 15 studies of programs that integrated treatment for a parent’s substance misuse and a parenting component. The primary focus of the review was on eight parenting outcomes including program retention, substance use, parenting stress, psychosocial adjustment, depression, child abuse potential, parenting practices, and parent–child interaction. The review found a lack of consistency between studies in the assessment instruments used. They found the mean retention rate for programs was 72%. Most studies reported a reduction in substance use. The review suggested that the important components to address in a supportive parenting program involve parenting stress, psychosocial adjustment, parental depression, the potential for child abuse, parenting practices, and parent–child interactions. The review found that studies that examined parent–child interaction produced improvements following engagement in the program. The review recommended that parenting interventions be routinely included in treatment programs for parents who misuse substances.

In conclusion, a range of trans-diagnostic therapies are now viewed as promising and evidence-based and are recommended for parents who misuse substances, while further research on the methodologies is required.

#### 3.12.7. vii–Clinician-Led Therapy

One form of therapy that has been used with clients who present with complex conditions involves delivery of both clinical services and in-home support, called clinician-led therapy. An example of clinician-led therapy was reported by Tustin [110] involving parents with severe mental health conditions who had co-morbidities of substance misuse, exposure to domestic violence, and difficulties in managing their child. The clinician-led program provided each parent with a combination of therapies delivered by one clinician together with in-home parenting education that was provided by a parenting coach who attended the family home and worked collaboratively with the clinician. The report does not provide information about the use of a standard outcome measure.

#### 3.12.8. viii–Reportable Therapy

Reportable therapy is a therapy that has a distinct confidentiality arrangement as the client requests a report about the efficacy of their treatment to be provided to an authority or a court. Reportable therapy is provided by a nominated primary clinician who undertakes to provide a treatment report to the authority at the end of therapy or at agreed times [111,112]. Clinicians who provide reportable therapy commonly follow a trans-diagnostic approach as they are required to address a range of presenting issues that are identified by the referring authority. Clients who receive reportable therapy might be the subject of a court order where they are required to participate in stated mandated activities.

Cashmore [113] made a call for improvements in the operations of courts in Australia that make orders mandating what parents must do. Cashmore expressed an opinion that the public expects courts to resolve complex problems such as parental substance abuse that social and health services have not been able to resolve, resulting in applications being made to courts to remove children as a protective measure before therapy interventions have been provided.

Cashmore noted that requiring courts to make decisions about the functioning of a family is traumatic for all members of a family. Both parents and children become defensive, and they are susceptible to the idea that they are being treated unfairly. Parents who feel anxious and helpless are prone to becoming argumentative and combative, reducing their cooperation with child welfare authorities. On the other hand, families who consider they are being treated fairly are more likely to engage in recommended therapy interventions.

Cashmore identified a range of steps that can be taken to foster a more collaborative and less adversarial approach between welfare and therapy services. She hypothesized that improved collaboration would lead to implementation of more effective interventions, improved child safety, better outcomes for families, and a reduced need to remove children from their homes while extended investigations occur. Cashmore also proposed that, when a parent views decision-making processes as fair, the parent is more likely to accept their own responsibility for negative outcomes that have occurred to a child.

Cashmore called for three improvements in the court process. The first call was for improvement in the quality of evidence provided to courts by expert assessors. Cashmore recommended that courts routinely provide feedback to assessors about the quality of their report and whether the court found their report helpful when deciding. A second topic for improvement involved feedback from courts to report writers and feedback to courts about the outcomes of their decisions regarding placements and the introduction of restrictive practices.

A third call by Cashmore sought the publication of de-identified decisions by courts. Cashmore noted there has been little research published about the effectiveness of interventions ordered by courts and about the long-term outcomes of recommendations by case workers. Cashmore noted that courts could receive feedback about the impacts of orders on the welfare of individual children, suggesting this would help judicial officers weigh the potential benefits and harm to children of the various alternative orders that a judicial officer can make.

Cashmore noted that legal decision-making is opaque rather than transparent due to an emphasis on privacy, as courts are closed, and many judgments are not published. Transcripts are accessible only if a party appeals a judgment, and few parents can afford to appeal decisions. Cashmore stated there are good arguments for Children’s Courts and other courts, including the Supreme Court that deals with adoption matters, to make the publication of de-identified final judgments, together with their reasons, standard practice.

#### 3.12.9. ix–Combined Pharmacotherapy and Psychotherapy

Ray et al. [114] reviewed 30 randomized clinical trials that examined the combination of cognitive behavioral therapy (CBT) and some form of pharmacotherapy for users of cocaine (23%), opioids (20%), and alcohol (15%). They found that the combined therapies produced effect sizes in the range 0.18 to 0.28.

#### 3.12.10. x–Integrated Multi-Disciplinary Clinics

One approach for treating people who misuse substances and who have multiple issues involves providing services from a multi-disciplinary team of clinicians who are co-located in an integrated clinic and operate from the same premises. Co-location is viewed as improving communication and collaboration between service providers.

Gwynne et al. [115] reported that three modes of early intervention had been shown to produce sustained improvements in children’s health, education, and wellbeing. They described a program that is provided in Australia called a Spilstead Model that delivers integrated care over a 12-month period through a community center that delivers the three approaches. Their evaluation found that the intervention produced changes in parent/child interaction by reducing parent stress and improving parental satisfaction, parent confidence, parenting capacity, family interactions, child wellbeing, and total family functioning. The study found that 71% of children who had presented developmental delays in initial screens functioned in the normal range on post-testing. Parents reported improvements in externalizing behaviors with a large effect size of d = 1.46.

Vidair et al. [116] reported an administrative arrangement that provided family-oriented services for families, as parents were routinely screened when they brought their distressed child to the clinic. The study found that 18.80% of mothers and 18.42% of fathers of distressed children themselves reported elevated internalizing symptoms. Vidair et al. reported that symptoms of parents were significantly associated with children’s internalizing and externalizing symptoms, and the arrangement permitted joint therapy to be provided. A similar opinion was reported by Middeldorp et al. [117].

Niccols et al. [118] reviewed 13 studies of impacts on children whose mother misused substances from interventions that were and were not integrated between health and welfare services. They found that infants obtained higher scores on developmental tasks when their mothers participated in integrated programs than when their mothers did not engage in treatment. Effect sizes up to 1.132 were found for children’s development, and effect sizes between 0.652 and 1.132 were found for children’s emotional and behavioral issues favoring integrated programs. Studies that compared integrated to non-integrated programs found that most improvements in emotional and behavioral functioning favored integrated programs with small effect sizes between 0.22 and 0.45.

A systematic review of studies aiming to address the three issues of parental symptoms, child symptoms, and parenting practices was provided by Everett et al. [119]. The review identified 25 psychotherapeutic interventions that directly intervened in parenting practices and that reported improvements in all three outcomes, but few interventions improved in samples where parents, children, or both met clinical-level thresholds of psychopathology.

In summary, one model of care for families where parents misuse substances involve the provision of services for both parents and children, using the resources of a multi-disciplinary team of collocated workers.

#### 3.12.11. xi–Continuing Case Management

Commentators have noted that addiction is a chronic disease where relapses occur frequently and recommend that people who are addicted to substances need to be viewed as having a chronic disease that requires continuing case management [120,121].

Morgenstern et al. [122] evaluated the efficacy of providing an intensive case management (ICM) service to coordinate long-term care for 302 substance-dependent women who were recruited from welfare offices that included referral to a substance abuse service. Follow-up occurred for 15 months. They reported that the ICM clients received a significantly higher level of service for their substance misuse, and they achieved higher rates of abstinence. However, due to the risks to children associated with frequent relapses when a person has an addiction, it appears wise that children do not remain in the care of a parent who is diagnosed with a relapsing addiction.

An alternative approach is to authorize supervision of a parent who is assessed as having a moderate level of substance misuse, while their child remains in the care of the parent. Legislation in some states authorizes courts to issue supervision orders, but these orders are rarely issued in Australia as child welfare departments rarely apply for a supervision order and courts lack mechanisms to monitor supervision orders ([1] p. 209).

#### 3.12.12. xii–Intensive Family Support

The use of intensive family support services to prevent the removal of children from parental care and to aid reunification has long been promoted [123].

The Australian Government introduced a voluntary Intensive Family Support Service (IFSS) program in 2016 for parents and caregivers of children aged 0–12 years where child neglect is a concern [124]. IFSS providers are expected to develop and maintain strong and productive working relationships with the local child protection authority which retains statutory responsibility for the ongoing case management, risk assessment, and risk management of the child. IFSS providers do not deliver specialist, clinical, or therapeutic interventions such as family counseling, financial counseling, or alcohol and drug treatment services. The IFSS scheme includes a referral process, a list of eligible funded activities, a needs assessment process, a support plan, an exit plan, and a workforce development strategy. Outcomes are to be measured, including using a Child Neglect Index [125]. Services are delivered by small teams who are locally employed professional and paraprofessional family support workers, who work under close professional supervision. The intensity of IFSS provision commences at 20 h per week and is scaled back as progress is made with each individual family. A caseload per worker between five and eight families across a year is proposed. An extension of service provision over 12 months can be negotiated.

The AIHW [23] subsequently defined a family support service as intensive if a family receives a service for an average of 4 h per week for a specified period, usually less than 180 days, and the purpose of the service is to prevent family separation or to reunify families following separation.

Forrester et al. [126] reported an evaluation of an intensive family support service that was provided to 279 children and compared outcomes to a control group. They found that 40% of children in both groups entered care, but children from families who received intensive family support took longer to enter care, spent less time in care, and were more likely to be at home on follow-up, and this option produced significant cost savings. Permanent changes were achieved for some families. Significant changes were not sustained for families who were assessed as having complex and long-standing problems, including sustained substance misuse.

Milligan et al. [127] studied the efficacy of programs to support mothers with substance abuse that were integrated between service delivery systems, were non-integrated, or where no intervention was provided. They found that the outcome measures of urine toxicology and the percentage of mothers using substances significantly favored integrated programs over no treatment. However, integrated programs were not significantly more effective than non-integrated programs in terms of outcome measures.

Al et al. [128] conducted a meta-analysis of the efficacy of brief, in-home intensive family preservation (IFS) programs that aimed to prevent out-of-home placement and improve family functioning, based on data from 20 studies including 31,369 participants. They found that, while IFS programs had a medium positive effect on family functioning for families with multiple problems (d = 0.486), the programs were generally not effective in preventing out-of-home placements.

Macvean and colleagues [129] were commissioned by the child welfare department of the State of New South Wales in Australia to review services funded by the child welfare department and provided by non-government agencies to assist families where children were vulnerable as their parents had been notified to the department. The purpose of the review was to identify interventions that were effective in improving outcomes for families with a range of identified vulnerabilities and to inform the reformation of service delivery. Macvean reviewed information about the efficacy of interventions provided by four authoritative international clearinghouses and rated 45 efficacies of interventions using the four categories of well-supported/supported/promising/emerging. The review group separated interventions according to risk factors into four categories: community, family, parent, and child. Exposure to family violence was categorized as a family risk factor. Parental substance misuse was categorized as a parent risk factor.

The Macvean review found that services funded by the child welfare department focused primarily on families in crisis where children were at risk of significant and immediate harm, rather than providing early preventive intervention. The group found that families in crisis commonly displayed many issues and that addressing one issue such as improving parenting skills often produced improvements for other outcomes such as maternal substance use and depression. The authors noted that the absence of agreed measures of outcomes between studies made it difficult to draw clear conclusions.

Macvean and colleagues discussed the applicability of interventions for families in specific circumstances. Macvean’s Table 6 identifies intervention programs for parental substance misuse and identifies one program as supported (Healthy Families America home visiting), one program as promising (Adult-Focused Family Behavior Therapy), and four programs as emerging (Early Start, Families Facing the Future, Family Connections, and Parents Under Pressure). In this review, Table 1 identified five programs for domestic violence, with one rated as promising and four rated as emerging.

Effective programs were discussed briefly by Macvean et al., including statements about qualifications required by providers. Program interventions could be delivered by an individual qualified clinician, a multi-disciplinary team, or a supervised worker. Some services were made available for 24 h of the day.

The Macvean group noted that programs commonly included many components. Macvean and colleagues identified components that were shared between all programs as including parenting education and a focus on parent–child relationships. The group noted there were common procedural steps such as assessment, but assessment instruments were not in shared use. Commonly, interventions were delivered in homes and were adjusted for individual needs; interventions involved weekly contact and continued for periods of up to 6 months; many interventions were delivered by trained staff; and efficacy was assessed after 6 months of intervention using diverse outcome measures.

The group identified gaps in services provided by the non-government agencies including the following: few focused interventions for families where domestic violence occurred and few focused interventions for families with co-existing substance misuse and mental health issues. Some interventions were available only for parents or only for children, rather than for parent–child pairs.

Macvean et al. reported the use of intensive case management, which they defined as providing intensive support by a case coordinator to people with high needs, with the aim of reducing high-risk behavior and increasing stability for a child. Intensive case management provides extended hours of service availability, after-hours crisis support, and outreach intervention and is managed by Intensive Family Support and Intensive Family Preservation services, which provide 24 h support over 12 weeks for families in crisis where children are at high or imminent risk of removal and placement in out-of-home care.

The Macvean review did not find that local service delivery agencies used any of the evidence-based interventions identified in their review. Nor did the Macvean review find that the funding agency required the use of evidence-based interventions. It is appropriate for policy makers to encourage service providers to use evidence-based interventions.

One policy approach that is used in Australia is for child welfare departments to allocate funding to non-government services to provide parenting education to families who experience multiple issues, where the non-government agency that provides support and education is not required to cooperate with therapy services that are provided to the family. It is proposed that care is needed either to ensure that workers who provide disability support to citizens are trained for the work they provide or for disability support workers to act in liaison with clinicians.

Bezeczky et al. [130] conducted a review of intensive family preservation services and concluded there is a need for an objective assessment instrument to distinguish families where parents can and cannot be rehabilitated, as the aim of preventing all out-of-home placements is not always appropriate as there are some family circumstances where children are better off being removed from parental care.

To assist the evaluation of intensive family support services, the South Australian Department of Human Services identified a list of instruments that can be used to evaluate programs, including thirty-five generic instruments for all population groups and eight instruments designed for use with the Aboriginal population. Advice for agencies on how to select a suitable measure to evaluate a program was provided by Goldsworthy and Robinson [131]. Policy makers could review methods to evaluate programs and publish methods they endorse.

One recommendation for policy is that all members of the workforce who provide intensive family services are qualified to meet the individual needs of their clients.

#### 3.12.13. xiii–Co-Located Workers

An initiative was reported by Zufferey et al. [132] with the aim of improving collaboration between two departments that support parents with comorbidities of substance misuse and mental health issues. The South Australian Government supported a Mental Health Liaison Project where a senior mental health nurse was relocated into a child welfare office to promote collaboration between the two departments that had traditionally adopted, respectively, either an adult focus or a child focus, rather than a family focus. The roles of the mental health nurse were to educate welfare workers and to facilitate referrals to appropriate specialist services.

An evaluation was conducted by Zufferey, who interviewed workers and parents. The evaluation found that placing a mental health nurse in a child welfare department enhanced the parent’s perception of empathy and respect shown by welfare officers. Parents had previously perceived the child welfare service as being a surveillance system, and they viewed the system as being less stigmatizing once the liaison project commenced.

## 4. Conclusions

This section identified a wide range of therapy interventions and practices in use to assist families where children are at risk of harm due to parental substance misuse. Interventions were categorized as suitable for families where children are at differing levels of risk of harm.

Five therapy interventions were identified as being indicated and focused interventions that are suitable as early intervention therapies for families where hazardous practices are emerging or are well established. The five early intervention therapies are therapy for one disorder, joint parent–child therapy, dual-focus therapy, trans-diagnostic therapy, and clinician-led therapy. It is proposed that these early intervention therapies are suitable for families who display a moderate level of risk factors and can be provided by a skilled clinician who receives appropriate support.

Reportable therapy is identified as a targeted intervention as it applies to a distinct cohort of people who are mandated by the court to participate in identified interventions.

Therapies for families that are provided by a multi-disciplinary team are classified as targeted interventions if the interventions are required by an authority. Three interventions that require funded input from many practitioners are classified as targeted, being services from a multi-disciplinary team, case management, and intensive family supports.

The review did not find evidence of efficacy for two interventions that are currently in use. One intervention involves referring citizens with multiple issues to multiple independent therapists who each treat one issue in an uncoordinated manner. A second intervention that lacks evidence of efficacy occurs when a child welfare worker has a dual role of being a counselor for one week and then being a prosecutor who prepares a case for court the next week. As noted by Justice Nyland, clients often lack trust in a practitioner who tries to combine the dual roles of being a therapist and a prosecutor (Nyland p. 196).

There is a need for ongoing research into the efficacy of the forms of therapy cited in this review and for policy to be developed involving the provision of early intervention therapy and support for families where children are vulnerable due to parental misuse of substances.

### 4.1. G–Privacy and Confidentiality

As discussed above, the child protection legislation CYPSA includes a mandatory notification clause requiring health professionals to inform the child welfare department of any reasonable suspicion that a child might be at risk of maltreatment. The mandatory notification impinges on the confidentiality of family information as people usually have a right to privacy and non-disclosure of their personal information. The topic of confidentiality is emphasized in codes of ethics of professional bodies, and health professionals are bound by strict confidentiality where they are prohibited from disclosing any personal information to third parties unless the disclosure is specifically authorized by the client involved or by legislation.

The topic of confidentiality is especially important when there is an assessment of increased potential for family violence. Clinicians who provide therapy to parents and children often ask the parents to sign a distinct restricted confidentiality contract that permits the clinicians to disclose and exchange information with all clients in the family at the discretion of the clinician.

If a person receives treatment from a multi-disciplinary team of clinicians, then it is conventional for one member of the team to be identified as a service coordinator, for the coordinator to prepare a treatment plan that lists inputs agreed upon by each team member where the treatment plan discusses confidentiality arrangements, and for the coordinator to distribute the treatment plan to all members of the treating team and to the client.

A policy system might be envisaged where a government introduces a whole government approach that includes the management of family violence and that requires the exchange of some information between participating therapists. This would require a distinct confidentiality arrangement for the disclosure of personal information.

The following four-part proposal is made if a whole government approach is introduced to manage family violence. First, strict confidentiality applies when one provider delivers indicated or focused services to a voluntary client. Second, if reportable therapy is arranged, then a referrer provides relevant information to the clinician who provides a treatment report to identified agencies, giving a restricted confidentiality contract. Third, members of a treating team who work with targeted groups are authorized to exchange information with each other about inputs they provide, progress made, and the need for their services to be ongoing. Fourth, if a person with multiple complex problems receives services from a range of independent providers, then a case manager is appointed who is responsible for maintaining suitable confidentiality.

The Australian Family Law Act defines the authority of parents to include a parent making decisions regarding children aged under 18 years about major topics that have long-term impacts on children, called parental responsibility. Section 60CC of the act discusses equal shared parental responsibility (where both parents have the same right to make decisions regarding their child) and sole parental responsibility (where the authority to make certain decisions is granted by a court order to one parent).

Family courts in Australia use distinct procedures when they ask community clinicians to provide therapy for vulnerable families, through a Lighthouse project. The Lighthouse project manages cases that involve risk relating to parental drug and alcohol misuse, family violence, mental health issues, and child abuse and neglect, using a Family DOORS screen. Cases with the highest levels of risk are referred to an Evatt List where the case is managed by a specialist court process. If a case is allocated to the Evatt list, then information is gathered from state courts, child welfare authorities, police, and other relevant bodies.

Section 150 [5] of the South Australian Children and Young People (Safety) Act 2017 authorizes departmental staff to require clinicians who provide therapy for a parent whose child has been placed in departmental care to provide a written report to the department answering specific questions, under a threat of punishment including imprisonment for a year for non-compliance. Information provided by the clinician to the department may be used in the department’s case in court proceedings.

In conclusion, this section recognizes that distinct confidentiality arrangements are required if a government introduces a whole government approach to manage family violence associated with parental substance misuse. A proposal is offered about confidentiality arrangements that could be introduced. There is a need for research into the implications of providing early intervention therapy for policies about the confidentiality of personal information.

### 4.2. H–Coordination of Services

This review cited studies finding that many parents who misuse substances display other risk factors for harming their children and are labeled as having multiple complex needs. Writers recommend that families with multiple complex needs be provided with distinctive forms of therapy and support before their children are removed from parental care.

Questions arise about how to coordinate support services for families with multiple complex needs. This paper noted the emergence of skilled clinicians who can provide joint parent–child therapy, dual-focus therapy, trans-disciplinary therapy, and clinician-led therapy for families with moderate levels of multiple complex needs. As each therapy is provided by one clinician, it is proposed that the clinician be responsible for coordinating the therapies they provide.

The situation is more involved when a family is assessed as presenting with multiple complex needs that require input from a range of separate professionals or agencies. Questions arise about how input from several providers might best be coordinated to meet the best interests of children while ensuring that children remain safe.

The topic of how to coordinate care for citizens with multiple needs arises with other cohorts including aged people. Discussions of coordination of care have been provided by several writers. Schultz et al. [133] discussed coordination of health care. Hillis et al. [134] discussed coordination for children with complex needs. Seckler et al. [135] discussed coordination in aged care. Olson et al. [136] discussed coordination of services for children and adolescents with serious emotional disorders who received wrap-around services. Khatri et al. [137] reviewed 56 studies regarding the coordination of a range of client health services for differing cohorts of clients.

This paper recommends that primary care clinicians who provide the service that meets a client’s main needs take on the role of case coordination where appropriate. A case coordinator has several roles that can be summarized as follows: (a) to make or arrange objective assessments of the client’s abilities and needs; (b) either to deliver necessary services or to identify a suitable provider for each need; (c) to prepare a written treatment plan that records the client’s needs, providers, and goals, and to distribute the plan to all involved personnel; (d) to clarify confidentiality principles for the case; (e) to motivate a client to engage in relevant services; (f) to prepare a relapse management plan if a person’s condition is in remission; and (g) to encourage a client to develop a supportive community-based network where relevant.

Some agencies use an administrative case management model where the coordination of services is provided by a person who does not provide any direct therapy or welfare service, so the coordinator is independent of service delivery and perhaps is impartial. An administrative case manager might act as a navigator who refers families to service providers, or an administrative case manager might manage the budget that funds providers, especially if a client has been assessed as having a cognitive disability and is incapable of managing the task of coordinating services.

The literature reviewed shows that a skilled clinician who follows a trans-diagnostic approach is potentially able to provide interventions that target each of the four issues in the cluster (substance misuse, parental mental health, family violence, and difficulty in managing a child’s behavior), when each issue occurs at a moderate level. Further issues such as difficulty in managing finances and managing tensions with extended families might be referred to an independent service. In this scenario, a skilled clinician who provides a service to meet a client’s most important needs might be asked to take on the additional role of being a case coordinator.

Whitcombe-Dobbs and Tarren-Sweeney [138] reviewed nine studies of programs that delivered intensive parenting support for maltreating parents. Their review concluded that maltreating parents are not homogeneous, and they found that no program they reviewed demonstrated efficacy in reducing all risk issues and all types of child maltreatment. They concluded there is a need for ongoing research and policy development on how to provide coordinated services for families with multiple complex needs.

A report by Kaspiew et al. [139] involving cases managed by Australia’s Family Courts recommended there is a need for both (a) specialized screening and assessment approaches when family violence and/or child abuse are identified; and (b) specific case management procedures to ensure that litigation is managed quickly, cost-effectively, and in a way that is consistent with the best interests of the child. This review concludes that there is a similar need for an assessment screen that identifies the specific needs of a family where parents misuse substances leaving children at risk of harm, where the screen facilitates the efficient coordination of services offered to each family.

This section summarizes coordination practices that can be used for each level of risk. The following proposals are made, referring to Table 1. A parent assessed as posing no-risk and low-risk can self-manage services by accepting responsibility for decisions they make about receiving services. A parent assessed as presenting a moderate risk can also be viewed as being capable of self-management of services as a voluntary client and can be held accountable for their decisions to accept or decline recommended services. Parents assessed as posing a high risk and/or who require multiple services can have service delivery coordinated by a designated case manager. This classification emphasizes the need for an instrument that makes objective assessments of the level of risk to a child.

If a child has been placed into statutory care, then the involvement of services is coordinated by a case manager employed by a relevant government department.

This review identifies scope for ongoing research into the topic of how services might best be coordinated when a parent misuses substances, especially when a parent is assessed as displaying multiple complex needs, and for policy implications.

## 5. Discussion

This paper reviewed information about parents who use substances and introduced a risk of harm to their children. The review identifies four major findings. First, parents who misuse substances are not a homogeneous group as parents present different risk issues and have differing needs. Research indicates that some parents have a single issue of misusing substances, while other parents display additional risk issues and are viewed as having multiple complex needs. Second, research indicates that a significant proportion of parents who misuse substances display a cluster of risk issues including mental health issues, difficulties in managing their child’s behavior, and domestic violence.

A third finding is that in Australia many children at risk of harm due to parental substance misuse first attract attention when they are referred to the child protection service. Reviewers find that while the staff of child protection services can refer vulnerable children for external therapy and support, this has not been a common practice as welfare workers prioritize their role of protecting children from harm by removing children from parental care.

A fourth finding is that a range of early intervention therapies for parents who misuse substances have been identified. However, research about the efficacy of these therapies is in an early stage of development as samples of parents who have received treatment are heterogeneous. The review found no consensus amongst researchers or practitioners about screening instruments for a range of relevant topics including the assessment of the level of risk of harm to children, the severity of substance use, parenting capacity and how to distinguish families who do and do not warrant early intervention therapy, which measures are agreed to assess the wellbeing and best interests of children, what objectives to set for therapy, and what outcome measures to use.

The review proposes that there is a need for an assessment model that is agreed upon between relevant disciplines and that identifies parental risk factors that predict an increased likelihood of a child being harmed by parents who misuse substances. The review further proposes there is a need to identify a screen instrument that provides objective and reliable assessments of risk factors identified in the risk model. The review recommends that an assessment screen identify parenting based on five administrative categories according to the need for services, where the categories are as follows: universal/indicated/focused/targeted/unfit parent. It is proposed that an adequate instrument will facilitate the referral of vulnerable families to appropriate interventions.

The review proposes that there is a need to identify a model of risk factors that are relevant for parents who misuse substances. Once a model has been identified, there is a need to develop a range of assessment instruments to address each theme. An instrument is required that objectively and reliably identifies specific early intervention services to address the moderate needs of each family, as well as to identify families where risk factors are too severe and expose children to an unacceptable risk of harm.

An adequate assessment instrument will facilitate discussion about how risk factors are weighed and prioritized to ensure that a suitable balance is achieved between providing early intervention therapy and ensuring the safety of children and to ensure that recommendations are actionable.

A suitable screening instrument will meet legal standards regarding the quality of evidence submitted by an expert witness to a court. This review proposes that a suitable screen will facilitate communication between therapists and legal services and will assist assessors who provide reports to courts. It is proposed that the introduction of a suitable screening instrument will significantly enhance the delivery of early intervention services to families where children are vulnerable to developing mental disorders and are at risk of being placed in out-of-home care.

The review identified a range of early intervention therapies that can be provided by skilled clinicians who deliver indicated and focused interventions. The review discussed mechanisms to coordinate services for parents who have multiple complex needs and proposed that a suitable screening instrument might contribute both to identifying parents with multiple complex needs and to decision-making about the coordination of services for these families.

## 6. Limitations

There are limitations to this review. Several references cited were funded by governments and are gray literature that might not have been peer-reviewed, introducing the possibility of bias. The review is a rapid review and does not provide a comprehensive investigation of many publications. As the topics reviewed are broad, the relevant literature is scattered over many publications. As there is not a long history of topics reviewed being discussed in academic journals, other reviewers might select different keywords when conducting their literature searches.

The review focuses on policy practices in one country Australia, and the field would benefit from further investigations that permit comparisons with practices and policies in other wealthy countries. International research on the topics addressed in this review is encouraged.

Decisions about whether to leave a child in a home where there are risk factors while parents engage in therapy or to remove a child from parental care raise significant ethical issues that are not discussed in this review. Ethical decisions about children at risk of harm are made by courts who are informed by evidence provided by expert witnesses. Based on Section 18ZH of the Family Law Act, Australia has followed a practice where family law courts appoint a single expert witness called a family consultant who provides assessments of all family matters affecting both the child and parent that must be addressed by the court, without the benefit of using instruments that are widely accepted. Australian family-oriented courts now accept treatment reports that are submitted by a clinician who has provided therapy to a family member, at the discretion of the court (Tustin, 2024) [112]. One policy innovation to reduce ethical concerns is for courts to appoint one expert to assess children and an independent expert to assess parents so the court receives two reports that address topics of concern to the two parties.

## 7. Future Directions

Several important topics and themes are identified in this review, and it can be argued that each topic requires additional research.

The review found that parents who misuse substances are not homogeneous as they have varying needs, and they present varying levels of risk of harm to their children. Future research can identify a scale that can be used by researchers and practitioners to assess the needs of parents who misuse substances, and the risk levels posed to children.

The paper reported studies from Australia that found some parents who misused substances had comorbidities of mental illness, exposure to family violence, and difficulty in managing their child’s behavior. Researchers might investigate whether the phenomenon of multiple complex needs occurs in countries other than Australia.

The paper proposes that the cluster of parental issues involving substance misuse, mental illness, exposure to family violence, and difficulty in managing children’s behavior occurs and can be managed by skilled clinicians who take an interest in this cohort and use a trans-diagnostic approach in therapy. This hypothesis warrants further research, including identifying components of effective transdiagnostic therapy for this cohort.

Further questions arise about whether clinicians are adequately trained in trans-diagnostic therapy for this cohort of parents and whether clinicians are trained to present objective reports to family-oriented courts.

The review proposes that if a coordinated whole government approach is to be achieved to manage cohorts of parents who misuse substances, then legal issues of confidentiality will need to be addressed, and confidentiality arrangements will need to vary according to risk categories. Topics around confidentiality need to be researched.

Finally, the paper addresses questions about how to coordinate service delivery for parents with multiple complex needs who are assessed as posing different levels of risk of harming their children. The review proposes a hierarchy where some parents self-manage service delivery, a skilled clinician is responsible for coordinating the delivery of some services as risk and complexity increase, and a case manager is appointed in some circumstances. The viability of this administrative arrangement needs to be researched.

## Figures and Tables

**Table 1 ijerph-22-00612-t001:** Levels of parental risk to children and recommended interventions.

Definitions of Levels of Intervention		
Level of Risk	Eligible Family	Type of Focused Intervention
No risk	Competent parent, has no risk factors	Universal educational and support interventions
Low risk	Vulnerable, parent presents a low risk of harm to child	Indicated Group psychoeducation Social support group
Moderate risk from one disorder	Vulnerable, parents have a diagnosed disorder and are eligible for indicated therapies provided by a clinician	Indicated/Focused Individual therapy for one disorder Joint parent–child therapy Dual-focus therapy Trans-diagnostic therapy for co-morbidities Clinician-led therapies
Moderate risk of cumulative harm	Vulnerable, parents present a moderate risk of harm and are willing to participate in voluntary individual therapy that is self-managed	Indicated/Focused Therapies as determined by an individual assessment
Moderate risk—Reportable therapy	Vulnerable, parents receive services coordinated by a key clinical who may provide a treatment report to an authority	Targeted Services are mandated by a court order and are identified by an individual assessment
High risk—Multiple complex conditions	Parents present with many risk factors	Targeted Therapy from multi-disciplinary clinic Case management Intensive family supports
Unacceptable risk	Unsafe, the child is removed from parental care and placed into statutory care	Targeted Out-of-home care

## Data Availability

No new data were created or analyzed in this study.

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
