# Peer review of "Review of Studies Regarding Assessment of Families Where Children Are at Risk of Harm Due to Parental Substance Misuse"

_ijerph, 2025, doi:10.3390/ijerph22040612_

Round 1
Reviewer 1 Report
Comments and Suggestions for Authors
The manuscript is interesting, but the methods section need a huge improvements. Here are the detail:
1. Abstract
The abstract does not follow the template guidance. The aim of study should be included in the background. Please use : to separate the structure. For example, Background: ...... Methods: ..... Results:....
2. Introduction
- Please put citation in paragraph 2.
- There is no strong evidence why this study is important due to lack of literature review. Please improve this part.
3. Method
- The method section is too weak and unable to elaborate all process. I recommend to add information which database used, queries and keywords used (in table), article inclusion criteria and selection, the diagram of selection process, etc.
Author Response
Response 1. I agree and have added aims in the Abstract.
Response 2. I agree and have added 2 citations in the second paragraph.
Response 3. Reviewer 2 made a similar criticism and provided detailed headings to use in Methods. I have followed the advice of Reviewer 2.
Reviewer 2 Report
Comments and Suggestions for Authors
Dear Authors,
Thank you for giving me the opportunity to review your manuscript. I found the paper to be highly valuable and well-written, and it certainly has the potential to make a significant contribution to the field. However, I do have one major concern that needs to be addressed: the method section.
In a review paper, the method section is particularly critical as it ensures transparency, reproducibility, and credibility. To strengthen this section, I recommend including the following key elements:
- Databases and Sources : Specify the databases, journals, or repositories you searched (e.g., PubMed, Scopus, Web of Science, Google Scholar).
- Keywords and Search Terms : Provide a list of the keywords, phrases, or Boolean operators used in your search queries. For example:
- "Socioeconomic status AND healthcare"
- Inclusion of Grey Literature : Clarify whether non-peer-reviewed sources, such as conference proceedings, dissertations, or reports, were included.
- Language and Date Restrictions : Indicate any limitations regarding language or publication dates.
- Inclusion Criteria : Define the characteristics of studies or articles included in your review (e.g., study type, population, intervention, outcomes).
- Exclusion Criteria : Specify the reasons for excluding certain studies (e.g., irrelevant topic, poor quality, lack of access to full text).
- Variables Extracted : List the data points collected from each study (e.g., authors, year, sample size, methodology, results, conclusions).
- Tools Used : Mention if spreadsheets, software, or other tools were used for data extraction.
- Consistency Checks : Explain how consistency in data extraction was ensured (e.g., pilot testing, inter-rater reliability checks).
- Ethical Considerations : Address any ethical considerations relevant to your review process.
Unfortunately, I could not find these elements clearly outlined in the current version of your manuscript. Please revise the method section to include these details, as they are essential for a robust review paper. Once these revisions are made, we can discuss the manuscript further.
Thank you for your attention to this matter, and I look forward to reviewing the updated version of your work.
Best regards,
Comments on the Quality of English LanguageThe language is acceptable.
Author Response
Response 1. I have added the type of search used.
Response 2. I have added key words used in searches in Methods.
Response 3. I acknowledge the literature includes grey literature in a Limitations section.
Response 4. I have added the restrictions about literature reviewed re language and dates in Methods.
Response 5. I have added inclusion criteria.
Response 6. I have added exclusion criteria.
Response 7. Themes identified and variables are now referred to throughout the paper.
Response 8. I now refer to assessment instruments identified throughout the paper.
Response 10. Ethical issues are addressed now in Discussion.
Reviewer 3 Report
Comments and Suggestions for Authors
The study addresses an important public health issue—parental substance misuse and its association with family violence and child welfare. The manuscript presents a comprehensive review of policies and empirical evidence regarding this topic, focusing on Australian child protection policies. The article is well-structured, providing a thorough discussion of risk factors, intervention strategies, and policy implications.
Recommendations:
- The discussion should more clearly differentiate between policy recommendations and empirical findings.
- There is limited discussion on potential biases in the reviewed studies, such as variations in how parental substance misuse is defined across different reports.
- A comparison with interventions in other high-income countries, such as the UK, the US, or Canada, would strengthen the study’s generalizability.
- Including more quantitative data on intervention outcomes (e.g., reduction in child maltreatment cases, improved parental compliance with treatment) would add depth to the analysis.
- A more structured discussion on how risk factors can be weighted or prioritized would make the recommendations more actionable.
Shorten lengthy paragraphs and ensure clear distinctions between findings, policy implications, and recommendations.
Author Response
Response 1. I have added clearer headings to differentiate policy and empirical findings.
Response 2. I have added recognition that grey literature is referred to with a possibility of bias being introduced. Lack of agreement about how substance misuse is defined is now acknowledged.
Response 3. I now acknowledge that information about differing policies and practices between countries is not addressed in this paper, and that further research on this topic is warranted.
Response 4. Available quantitative data about efficacy of treatments is now provided.
Response 5. The topic of using an assessment instrument to encourage structured discussion about weighing and prioritizing risk factors and to make recommendations actionable is now included about line 1732.
I have now used the language of distinguishing between findings, policy implications and recommendations throughout the paper.
Round 2
Reviewer 1 Report
Comments and Suggestions for Authors
The manuscript has been improved, but the writing should be improved:
- There are so many short paragraphs consist of a single sentence, for instance: line 68, 93, 102, 108 and many more. Please develop the paragraph by providing more elaboration. a paragraph at least 3 sentences.
Author Response
The short paragraphs have been identified and elaborated. There are now few single sentence paragraphs.
Reviewer 3 Report
Comments and Suggestions for Authors
The revisions and response to the authors are acceptable
Author Response
The reviewer reported that responses to round 1 are acceptable.